# Loss of dyskerin facilitates the acquisition of metastatic traits by altering the mevalonate pathway

Evelyn Andrades[1], Agustí Toll[1,2], Gustavo Deza[3], Sonia Segura[3], Ramón Gimeno[4], Guadalupe Espadas[5,6], Eduard Sabidó[5,6], Noemí Haro[7], Óscar J Pozo[7], Marta Bódalo[8], Paloma Torres[1], Ramon M Pujol[3], Inmaculada Hernández-Muñoz[1]

The initial dissemination of cancer cells from many primary tumors implies intravasation to lymphatic nodes or blood vessels. To investigate the mechanisms involved, we analyzed the expression of small non-coding RNAs in cutaneous squamous cell carcinoma (cSCC), a prevalent tumor that mainly spreads to lymph nodes. We report the reduced expression of small nucleolar RNAs in primary cSCCs that metastasized when compared to non-metastasizing cSCCs, and the progressive loss of DKC1 (dyskerin, which stabilizes the small nucleolar RNAs) along the metastasis. DKC1 depletion in cSCC cells triggered lipid metabolism by altering the mevalonate pathway and the acquisition of metastatic traits. Treatment of DKC1-depleted cells with simvastatin, an inhibitor of the mevalonate pathway, blocked the expression of proteins involved in the epithelial-to-mesenchymal transition. Consistently, the expression of the enzyme 3-hydroxy-3-methylglutaryl-CoA synthase 1 was associated with pathological features of high metastatic risk in cSCC patients. Our data underpin the relevance of the mevalonate metabolism in metastatic dissemination and pave the possible incorporation of therapeutic approaches among the antineoplastic drugs used in routine patient care.

## Introduction

The metastatic spread begins with the detachment of the tumor cell, either alone or collectively, from the primary tumor to intravasate into lymph nodes or blood vessels, which are hostile environments in which most tumor cells are unable to survive (Lambert et al, 2017). In many cancers, including melanoma and breast cancer, regional lymph node involvement is associated with an increased risk of distant metastasis and worse outcomes (Ferris et al, 2012). Indeed, it appears that the initial stages of metastasis are more likely to occur through lymphatics in most solid tumors, although this dissemination route has been poorly characterized, especially when compared to the efforts invested in elucidating the mechanisms responsible for tumor spread through blood vessels to distant sites (Ubellacker & Morrison, 2019).

Cutaneous squamous cell carcinomas (cSCCs) constitute an ideal model to investigate general mechanisms for tumor dissemination, because samples from every stage in the cancer continuum are easily accessible during clinical practice. cSCC is a common neoplasia, and after basal cell carcinoma, it represents the second most prevalent malignant skin tumor. In many cases, it does not have an aggressive evolution, although ~4% of patients will develop nodal metastases and 1.5% dies as a result of this disease. Its high prevalence gives rise to a high mortality (more than twice as many as those killed by malignant melanoma). Overall, the survival of untreated metastatic cSCC is less than 35% at 5 yr. From 15% to 38% of the patients with high-risk cSCC have in-transit metastases (metastatic foci located between the primary tumor and the regional lymph node regions), and tumor cells can spread along lymphatic vessels and/or nerves. Analogous to in-transit metastases found in melanoma, their presence is a poor prognostic indicator in cSCC.

The acquisition of the metastatic capacity by the tumor cell has been the subject of extensive and numerous studies (for a recent review, see Massagué and Ganesh [2021]). The hypothesis that postulates that this competence is acquired through late genetic alterations throughout tumor progression has lost strength, and it is currently accepted that the cell with metastatic potential is already present in the early phases of tumorigenesis (Lambert et al, 2017). Malignant cells may have a high cellular plasticity that allows the adaptation to the different adverse conditions throughout the process of tumor spread. Alternatively, there might be a selection process that would favor the dissemination and survival of cells

[1]Group of Inflammatory and Neoplastic Dermatological Diseases, IMIM (Hospital del Mar Medical Research Institute), Barcelona, Spain   [2]Department of Dermatology, Hospital Clínic de Barcelona, University of Barcelona and Institut d'Investigacions Biomèdiques August Pi i Sunyer; Centro de Investigación Biomédica en Red de Enfermedades Raras, Instituto de Salud Carlos III, Barcelona, Spain   [3]Department of Dermatology, Hospital del Mar, Parc de Salut Mar, Barcelona, Spain   [4]Laboratory of Immunology, Department of Pathology, Hospital del Mar, Parc de Salut Mar, Barcelona, Spain   [5]Proteomics Unit, Centre de Regulació Genòmica, Barcelona Institute of Science and Technology, Barcelona, Spain   [6]Universitat Pompeu Fabra, Barcelona, Spain   [7]Applied Metabolomics Research Group, IMIM (Hospital del Mar Medical Research Institute), Barcelona, Spain   [8]MARGenomics, IMIM (Hospital del Mar Medical Research Institute), Barcelona, Spain

Correspondence: mhernandez@imim.es

present in the primary tumor that have already achieved this competence (Lambert et al, 2017). In any case, the molecular bases of metastasis are fundamentally epigenetic, because they allow the tumor cells to undergo specific, rapid, and reversible functional changes by reprogramming their transcriptome.

The general mechanisms through which the different transcriptional programs are modulated during the metastatic progression remain largely unknown. The identification of these mechanisms is currently an important scientific challenge, because the observed cellular changes may correspond to transient responses of the tumor cell to specific environmental conditions, and therefore difficult to model experimentally. We took advantage of the availability of primary cSCCs that had spread to lymphatic vessels to investigate the expression of the epigenetic regulators small non-coding RNAs (sncRNAs). These molecules can modulate the gene expression of a very broad set of targets, and their aberrant expression is linked to cancer development and progression and affects several processes such as proliferation, apoptosis, differentiation, and invasiveness (Zhang et al, 2021). To date, most of the sncRNAs identified that are involved in SCC progression correspond to the subclass of miRNAs, but studies aimed to determine the role of other sncRNA species are scarce. In this work, we show that primary metastasizing cSCCs (MSCCs) displayed a global impoverishment in small nucleolar RNAs (snoRNAs) when compared to non-metastasizing cSCCs (NMSCCs). Accordingly, the expression of dyskerin (DKC1), the protein that provides snoRNA stability, is reduced in cSCC metastases. Moreover, our proteomic, transcriptomic, metabolomic, and functional data revealed that DKC1 down-regulation results in a switch to the mevalonate pathway, potentially increasing cholesterol biosynthesis, and the acquisition of metastatic traits in cSCC cells. Quantitative scoring of the expression of the lipid enzyme 3-hydroxy-3-methylglutaryl-CoA synthase 1 (HMGCS1) in primary cSCC tumors and in metastatic samples underlined the relevance of the in vitro data and their translational potential for the clinical management of cSCC patients.

## Results

### Identification of sncRNAs differentially expressed in primary MSCC

sncRNAs are molecules with less than 200 nucleotides and include a heterogeneous group of RNA species such as miRNAs (that repress translation of target RNAs by directing them to the RNA-induced silencing complex) or the snoRNAs (Zhang et al, 2021). To determine whether sncRNA molecules are aberrantly expressed in MSCC, tumors were macrodissected from paraffin-preserved blocks after examination of hematoxylin–eosin-stained sections such that all samples displayed a minimum 70% enrichment for tumor or dysplastic cells. Analysis of sncRNA expression in 22 primary cSCC (10 from cSCCs that had evolved to histologically confirmed metastases, or MSCC, and 12 from a cSCC control group who had not developed any metastasis in a 5-yr follow-up period, or NMSCC) was performed with the miRNA 4.0 array (Applied Biosystems). Unsupervised hierarchical clustering of the most variably expressed

sncRNAs stratified the tumors into two groups, one formed mainly by NMSCC and the other formed exclusively by metastasizing cSCC, the latter displaying an overall sncRNA down-regulation (Fig 1A, left panel). Supervised clustering of MSCC versus NMSCC identified the differential expression of 330 sncRNAs, most of them down-regulated in the metastasizing group (248 sncRNAs; 75%) (Table S1). Functional and sequence-based classification of the probes in the array revealed alterations in the proportion of the snoRNAs represented in MSCCs when compared to their expression in NMSCCs (Fig 1A, right panel).

There are two main classes of snoRNAs, box C/D snoRNA and box H/ACA snoRNA, and a third less represented class, the small Cajal body–specific RNAs (scaRNAs) (Deogharia & Majumder, 2019). Among the snoRNAs differentially expressed in cSCC, up to 53 H/ACA snoRNA species were found down-regulated in MSCCs. The H/ACA box snoRNAs are molecules of about 120–250 nucleotides characterized by a structure with two hairpins connected by a region with an H box (Romano et al, 2017). Expression analysis by qRT–PCR of some down-regulated H/ACA snoRNAs confirmed the low levels of these snoRNAs in primary MSCCs (Fig 1B). Independent validation of these transcripts in a different cohort consisting of 47 biopsy samples confirmed snoRNA down-regulation in MSCC when compared to those cSCC that did not further progress (Fig 1C).

### DKC1 expression is reduced along cSCC progression

Most *H/ACA snoRNA* genes are located within intronic regions of protein-coding genes, lacking independent transcriptional regulatory elements (Dieci et al, 2009). Because MSCCs displayed a global perturbation in H/ACA snoRNA expression levels, we hypothesized that their down-regulation could be due to post-translational mechanisms. Human H/ACA snoRNAs are processed from the host gene introns by exonucleases, and their association with core proteins of the H/ACA RNP complex such as DKC1 is critical for their stability (Mcmahon et al, 2015). DKC1 is an RNA pseudouridine synthase that requires a single snoRNA molecule and auxiliary proteins for RNA substrate recognition by the H/ACA RNP complexes. To investigate the role of DKC1 in snoRNA expression levels, we initially chose the SCC13 cell line, originated from a well-differentiated human SCC of the facial epidermis (Rheinwald & Beckett, 1981). Transient transfection of these cells with a small interference RNA (siRNA) directed to DKC1 (siDKC1) showed that DKC1 knockdown resulted in drops in H/ACA snoRNA levels (Fig 1D). This effect was also observed in the highly differentiated cell line SCC12 and in the UT-SCC12A cell line, the latter established from a primary human cSCC that subsequently metastasized (Jääskelä-Saari et al, 1997) (Fig S1). These findings were consistent with previous data showing that loss of DKC1 reduces the accumulation of a subset of H/ACA snoRNA-derived miRNAs in different cellular contexts (Alawi & Lin, 2010) and supported the possibility that down-regulation of DKC1 underlies the impoverishment of H/ACA snoRNAs in metastatic cSCC. Therefore, we investigated DKC1 expression by immunohistochemistry in tissue microarrays of primary tumors and in metastases. This approach revealed that the number of MSCC displaying nuclear DKC1 staining tends to be lower than those that did not progress (63% versus 75%, respectively). In addition, the number of cSCC metastases expressing DKC1 was

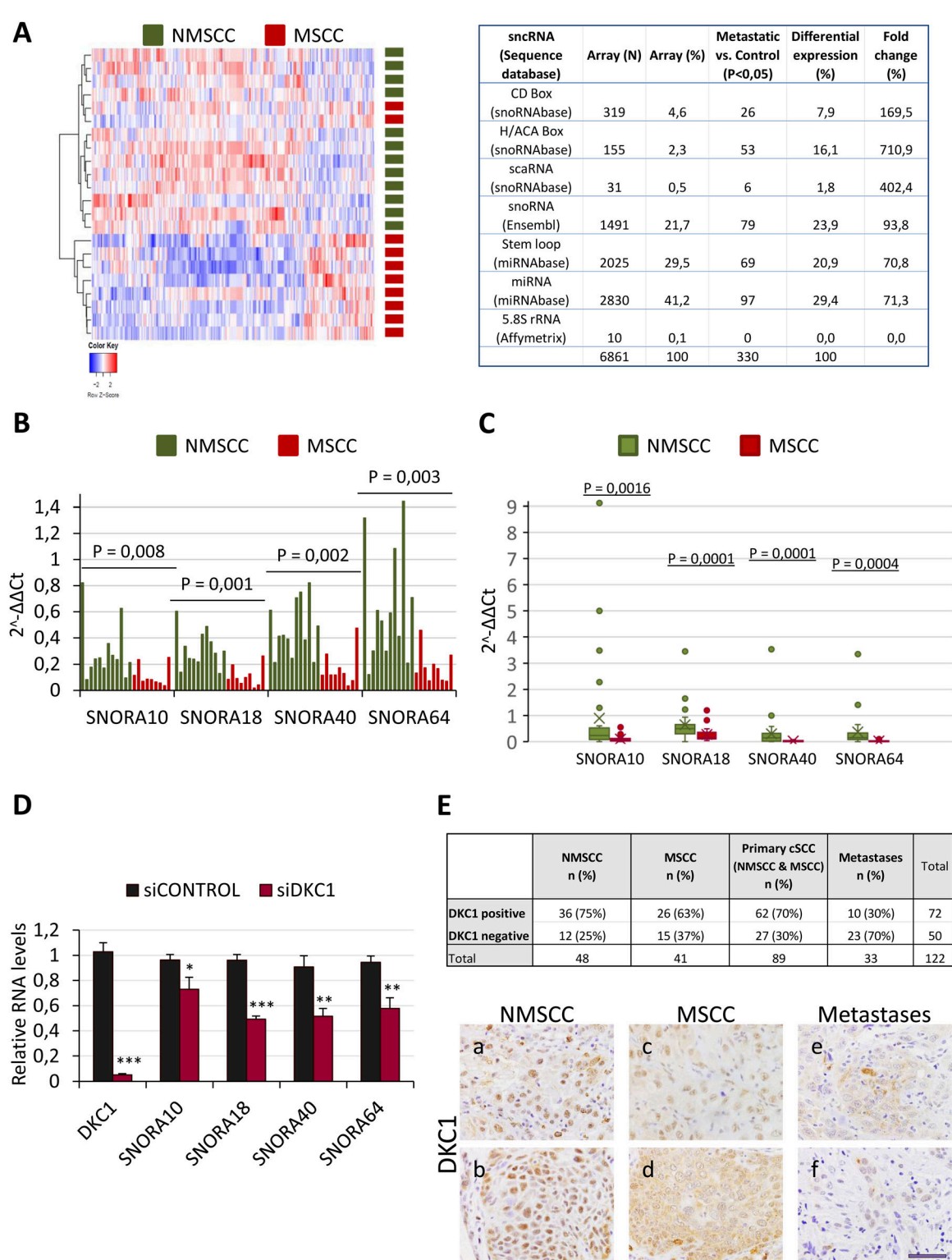

**Figure 1. snoRNAs are differentially expressed in MSCCs versus NMSCCs.**

**(A)** Unsupervised clustering of sncRNA expression arrays of cSCCs (left) and table summarizing the representation of different sncRNA species in the miRNA 4.0 array (Applied Biosystems) and the number and percentage of the sncRNA species differentially expressed in MSCCs when compared to NMSCC (right). N, number of species present in the array; %, relative representation of the species in the array. Fold change, changes in relative representation of the species differentially expressed in MSCC in reference to their representation in the array. **(B)** Expression of selected snoRNAs in NMSCCs (green) and MSCCs (red), analyzed by qRT–PCR and referred to U48 expression levels. **(C)** Expression of the selected snoRNAs in an independent cohort of NMSCCs (green) and MSCCs (red), analyzed by qRT–PCR and normalized to U48 control levels. In (B, C), the Mann–Whitney U test was applied to determine statistical significance. **(D)** DKC1 silencing in SCC13 cells reduces snoRNA levels, detected by

lower than DKC1-positive primary tumors (30% versus 70%, respectively; Fisher's exact test, *P* = 0.0001), confirming that DKC1 expression is further reduced along the metastatic progression (Fig 1E).

To determine whether DKC1 down-regulation could be related to the progression of other tumoral processes, we analyzed its expression in different cancer types using The Cancer Genome Atlas database. Elevated DKC1 expression in other tumor types such as lung adenocarcinoma or head and neck SCC (HNSCC) correlated with disease progression, according to previous studies (Angrisani et al, 2014; Kan et al, 2021) (Fig S2A). However, in lung SCC and ovarian cancer low DKC1 expression levels were associated with poor survival rates (Fig S2A) and had been also reported in endometrial cancer (Alnafakh et al, 2021). Furthermore, low expression levels of the other RNP core protein components NHP2, NOP10, and GAR1 were also associated with poor prognosis levels in lung SCC and ovarian cancer, but not in lung adenocarcinoma or HNSCC (Fig S2A). These observations suggest the notion that the expression of components of the DKC1-containing RNP complexes is coordinately down-regulated along the metastatic process in a specific subset of cancers.

### DKC1 depletion in cSCC triggers the activation of the mevalonate pathway

*DKC1* is among the 10% of the genes considered to be common essential genes, and complete abrogation of their expression in long-term cultures results in cell death in most normal and cancer cells (DepMap, 2022: The Cancer Dependency Map Project at Broad Institute). We therefore investigated the functions affected by DKC1 down-regulation by transient transfection of UT-SCC12A cells with siDKC1. Because DKC1 could regulate different biological functions by transcriptional and post-transcriptional mechanisms, we analyzed the effects of DKC1 down-regulation at both transcriptomic and proteomic levels. Intersection of differentially overexpressed transcripts and proteins in siDKC1 cells identified 82 genes (Table S2). Gene ontology analysis performed with the public database STRING revealed that this gene set was enriched in proteins involved in small molecule metabolic processes (Fig 2A). Indeed, among them, 42 had catalytic activity and 45 were involved in acetylation processes. Enhanced expression of some of these enzymes in UT-SCC12A cells with reduced DKC1 levels was confirmed by qRT–PCR (Fig 2B) and was also observed in other cSCC cell lines upon DKC1 depletion (Fig 2C). Among these proteins were the enzyme ACSS2 (acyl-CoA synthetase short-chain family member 2), which catalyzes the activation of acetate and produces acetyl-coenzyme A (acetyl-CoA) for use in energy generation and lipid synthesis, and HMGCS1, which facilitates the condensation of acetyl-CoA with acetoacetyl-CoA to form HMG-CoA, which is converted by HMG-CoA reductase (HMGCR) into mevalonate, a precursor for cholesterol synthesis. Enrichments in the expression

of these enzymes could respond to changes in the metabolic functions of the mitochondrial matrix. Oxidation of carbohydrates and fats in the tricarboxylic acid (TCA) cycle, initiated by the conversion of pyruvate and fatty acids to acetyl-CoA, is one of the most important functions carried out in the matrix. To investigate the steady state of the metabolic cycle in DKC1-depleted cells without technical interferences from the transfection procedure, we generated a UT-SCC12A stable cell line with the doxycycline-inducible expression of a DKC1 short hairpin RNA (shRNA). This approach had been previously used and proven effective in other cell types (Di Maio et al, 2017). In parallel, we also generated a UT-SCC12A cell line with the inducible expression of a control shRNA, to rule out potential doxycycline-induced effects. Detection of TCA analytes by LC–MS/MS showed reduced levels of the TCA intermediates in DKC1-depleted cells. Furthermore, lactate levels, produced by glucose degradation to pyruvate to generate ATP in cancer cells (the Warburg effect), were reduced up to 40% in DKC1-depleted cells (Fig 2D), indicating that their altered metabolism relies on energy sources other than glucose. Acetoacetate (substrate of HMGCS1 and precursor of mevalonate) levels were found to be also reduced in DKC1-depleted cells (Fig 2D), supporting the activation of the mevalonate pathway.

The process of energy metabolism involving lipid-derived molecules yields more energy than carbohydrates. The dependency of cancer cells on lipid metabolism can rely on the endogenous synthesis of the fatty acids or on lipid uptake through diffusion or by receptor-mediated endocytosis (Munir et al, 2019). Among the up-regulated proteins identified by LC–MS/MS in siDKC1 cells was the LDL receptor (LDLR), which mediates the endocytosis of cholesterol-rich LDL. As Fig 2E shows, inducible shDKC1 cells cultured in the presence of doxycycline displayed higher LDLR protein levels than those cells cultured in doxycycline-free medium and shControl cells. In contrast to LDLR, IGF1R levels were not affected by DKC1 depletion, indicative of DKC1 specificity in the regulation of lipid metabolism (Fig 2E). In addition, because the activation of the EGFR pathway has been previously linked to altered metabolic processes, including the "lipogenic phenotype" in glioblastoma multiforme (Guo et al, 2011), we investigated the effect of EGF treatment in these cells. However, high LDLR levels in DKC1-depleted cells were not affected by EGF treatment: although EGF treatment increased LDLR levels in UT-SCC12A cells expressing DKC1, it did not further enhance LDLR expression in DKC1-depleted cells (Fig 2E). Furthermore, LDL treatment reduced LDLR levels in control cells, according to the negative regulatory feedback loop by which LDL suppresses LDLR synthesis to limit LDL uptake, but not in DKC1-depleted cells (Fig 2E). These results suggested an EGF- and LDL-independent up-regulation of LDLR in DKC1-depleted cells.

The expression of *LDLR* and other genes involved in fatty acid and cholesterol metabolism such as *ACSS2*, *ACAT*, *HMGCS1*, or *IDI1*, all of them overexpressed in DKC1-depleted cells, is transcriptionally regulated by the sterol regulatory element–binding proteins (SREBPs

---

qRT–PCR. Statistical analysis was performed by the *t* test, *P < 0.05; **P < 0.005; and ***P < 0.001. **(E)** Table summarizing immunohistochemical expression of DKC1 in tissue microarrays of primary non-metastasizing and metastasizing cutaneous squamous cell carcinomas (NMSCC and MSCC, respectively), and in metastases. Initial cohorts: 56 MSCC, 51 non-MSCC, and 39 metastases. In some cases, signal could not be evaluated because of technical limitations. Below, representative images showing DKC1 immunodetection in independent tissue microarray samples of NMSCC (a, b), MSCC (c, d), or lymph metastases (e, f). Scale bar, 50 $\mu$m.

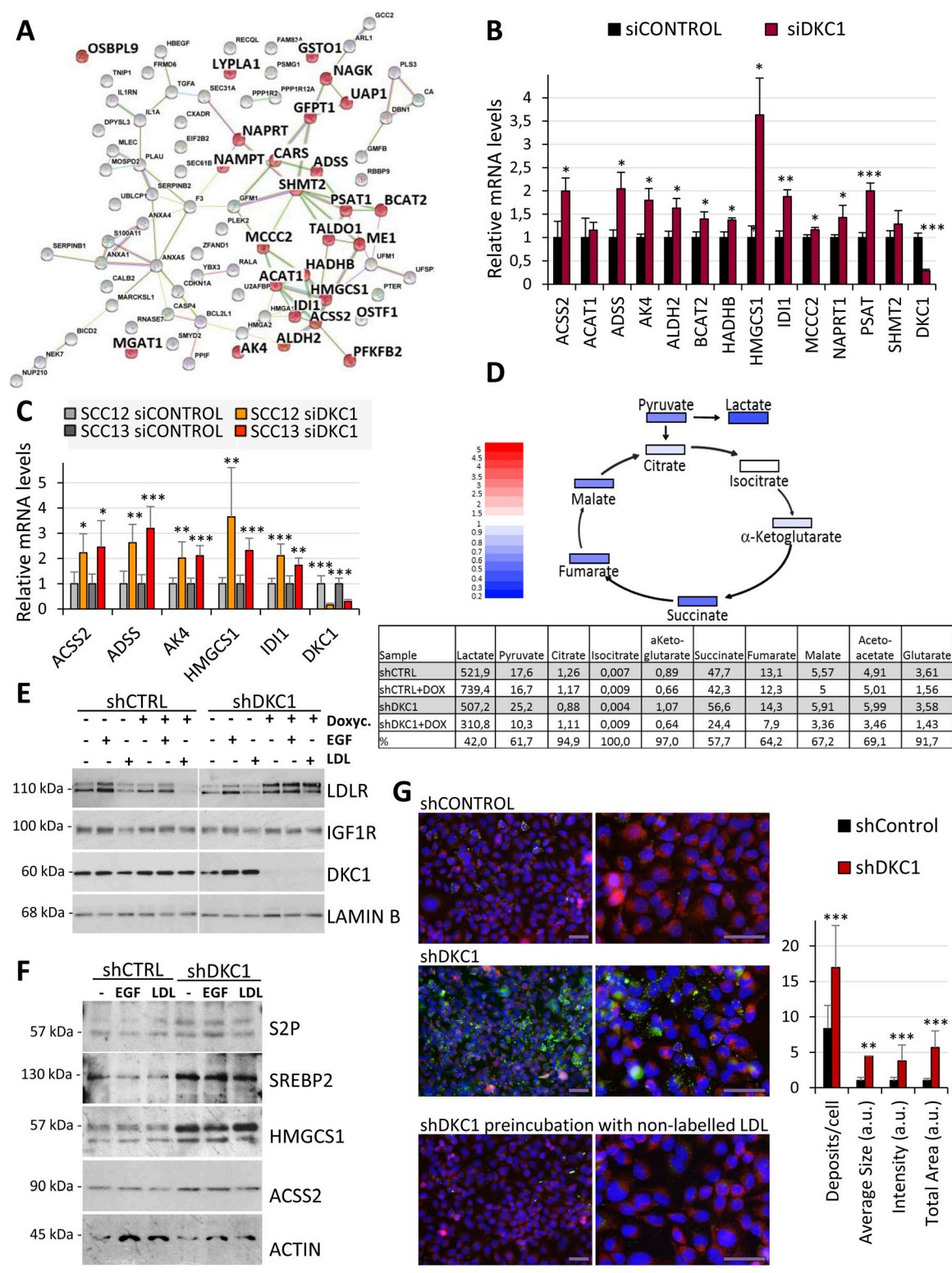

Figure 2.   DKC1 silencing triggers the activation of the lipid metabolism in cSCC cells.
(A) Gene interaction network derived from STRING database analysis of genes with enhanced mRNA and protein expression in UT-SCC12A cells transiently transfected with siDKC1 or control siRNA, determined by gene expression array and proteomic analyses. Genes labeled in red correspond to the gene ontology categories "Small molecule metabolic process, Nucleobase-containing small molecule metabolic process, Carboxylic acid metabolic process, and Coenzyme metabolic process." (B) Gene expression of selected enzymes in UT-SCC12A cells transiently transfected with control or DKC1 siRNAs, determined by qRT–PCR. (C) Expression of metabolic enzymes in SCC12 and SCC13 cells with reduced DKC1 levels. Results from two independent experiments performed in triplicate. For (B, C), statistical analysis was performed by the t test, *$P < 0.05$; **$P < 0.005$; and ***$P < 0.001$. (D) Upper panel, schematic representation of differences in TCA metabolites in UT-SCC12A cells that grew with doxycycline,

(or SREBFs)). SREBPs are transmembrane proteins that, in low cholesterol concentrations, are sequentially cleaved by S1P and S2P, and their N-terminal fragments are imported to the nucleus to activate transcription. SREBP2 mainly activates synthesis and uptake of cholesterol, whereas SREBP1 activates lipogenesis (Goldstein et al, 2006). To determine whether the lipid metabolism in DKC1-depleted cells was regulated by this mechanism, we analyzed the levels of the protease S2P, the transcription factor SREBP2, and acetyl-CoA metabolic enzymes. DKC1-depleted cells displayed enhanced SREBP2, S2P, ACSS2, and HMGCS1 levels when compared to their control counterparts, and EGF or LDL treatment did not affect the levels of any of these proteins (Fig 2F). Importantly, DKC1 silencing in lung SCC cells also resulted in the enhanced expression of proteins involved in lipid synthesis and energy generation (Fig S2B), suggesting that modulation of DKC1 levels could rewire the metabolism in different cancer cell types.

To investigate whether the enhanced LDLR levels functionally affected lipid uptake, we incubated control and DKC1-silenced cells with Alexa Fluor 488–conjugated LDL. As shown in Fig 2G, control cells barely incorporated LDL, whereas this lipoprotein was efficiently internalized in DKC1-depleted cells, supporting the notion that DKC1 down-regulation induces a switch from carbohydrate- to lipid-dependent metabolism. In summary, DKC1 down-regulation resulted in the expression of key proteins involved in the metabolism of the acetyl-CoA (ACSS2 and ACAT1) toward the biosynthesis of mevalonate-derived metabolites (i.e., HMGCS1 and IDI1), and increased cholesterol uptake (LDLR) to enhance isoprenoid and sterol levels.

### DKC1 down-regulation enhances the metastatic and stemness traits of cSCC cells

The up-regulation of the lipid metabolism is a key feature of cancer stem cells (CSCs) in breast cancer and oral squamous cell carcinomas, among other malignancies (Luo et al, 2017). Indeed, the mevalonate precursor enzyme HMGCS1 is a marker of CSC enrichment in breast cancer (Biddle et al, 2011), whereas CD36, the receptor for the palmitic acid, is overexpressed in CD44-positive metastasis-initiating cells in oral squamous cell carcinoma (Pascual et al, 2017). One of the most-accepted CSC surface markers is CD44, a glycoprotein receptor that triggers SRC–YAP–ZEB1 axis in cSCC cells to promote the expression of a mesenchymal program (Pastushenko et al, 2021), which has been associated with an increased metastatic risk in cSCCs (Toll et al, 2013). We therefore investigated whether DKC1 down-regulation affected the expression of proteins that could be involved in cSCC metastasis. DKC1 silencing enhanced CD44 and YAP1 expression in the poorly differentiated UT-SCC12A cells, but not in the well-differentiated SCC12 or SCC13 cells (Fig 3A), suggesting that DKC1 depletion could favor

the acquisition of metastatic traits in cSCC cells within a specific dedifferentiation status or in a hybrid EMT state (Pastushenko et al, 2021). To further investigate this possibility, UT-SCC12A cells with inducible shDKC1 expression were cultured in the presence of doxycycline. As shown in Fig 3B, DKC1-depleted cells displayed enhanced CD44 and TWIST levels when compared to control cells. Interestingly, EGF and LDL treatments induced the expression of both proteins in control cells, whereas their expression remained high regardless of the treatment in DKC1-depleted cells. In contrast, RING1B expression, which we have previously reported to be directly associated with a poor prognostic in cSCC, was not affected (Fig 3B) (Hernández-Ruiz et al, 2018). According to the acquisition of an EMT phenotype, DKC1-depleted cells exhibited morphological changes and tend to assume round-shaped or elongated-fibroblastoid morphologies (Fig 3C). To evaluate whether these phenotypic changes functionally affect cell motility, migration and invasion were subsequently determined. These assays showed that DKC1-depleted cells displayed a greater ability to migrate and invade in vitro (Fig 3D).

Lipid metabolism and EMT have been previously associated with tumor stemness and increased clonogenicity in vitro (Nieto et al, 2016; Latil et al, 2017; Pastushenko & Blanpain, 2019). To test whether DKC1 also affected clonogenicity, we performed 3D tumor spheroid assays. DKC1-depleted cells generated larger aggregates than control cells, indicative of their enhanced abilities to self-renew and to generate daughter cells (Fig 3E).

### Enhanced mevalonate pathway in DKC1-depleted cells regulates their metastatic traits

To understand how DKC1 down-regulation progressively affects the expression of proteins regulating lipid metabolism and metastasis in cSCC cells, we performed a time-course experiment. RNA and protein from cells with inducible shControl or shDKC1 expression cultured with doxycycline up to 5 d were extracted. Of note, maximal silencing of DKC1 mRNA expression was reached within ~24 h of doxycycline treatment, whereas complete depletion at the protein level occurred at day 3 (Fig 4A and B). qRT–PCR analysis showed that the expression of LDLR and enzymes involved in the mevalonate pathway was progressively induced in the first 2 d of doxycycline treatment, and further increased or was maintained at day 5 (Fig 4A). CD44 expression peaked at day 2, whereas SNAIL, TWIST, ZEB2, and VIMENTIN reached maximum levels at days 4–5 (Fig 4A). These increases at the transcriptional level were reflected by an enhanced expression at the protein level (Fig 4B), confirming that DKC1 down-regulation induces transcriptional changes in lipid metabolism and the acquisition of metastatic traits in cSCC cells.

To better dissect the relationship between lipid metabolism and EMT, cells were treated with simvastatin, a potent HMGCR inhibitor

---

expressing shDKC1 or control shRNA. Blue coloration represents down-regulation in DKC1-depleted cells. Lower panel, numeric differences in the levels of TCA cycle metabolites in shControl and shDKC1 cells that grew in the absence or presence of doxycycline, analyzed by gas chromatography–mass spectroscopy. **(E)** LDLR and IGF1R protein levels in control and DKC1-depleted UT-SCC12A cells in basal conditions or treated with EGF or LDL. LAMIN B, loading control. **(F)** Levels of proteins involved in lipid homeostasis in control and DKC1-depleted UT-SCC12A cells cultured with doxycycline in basal conditions or treated with EGF or LDL. S2P, site-2 protease; SREBP2, sterol regulatory element–binding protein 2; HMGCS1, 3-hydroxy-3-methylglutaryl-CoA synthase 1; and ACSS2, acyl-CoA synthetase short-chain family member 2. ACTIN, loading control. **(G)** LDL uptake in control and DKC1-depleted UT-SCC12A cells cultured with doxycycline for 3 d in basal conditions, analyzed with BODIPY-labeled LDL (green) upon costaining with LC3 (red) and DAPI (blue), and detected by fluorescence microscopy as shown in the representative images. Scale bars, 50 $\mu$m. On the right, quantification of LDL deposits analyzed with ImageJ. The t test, **P < 0.005 and ***P < 0.001.

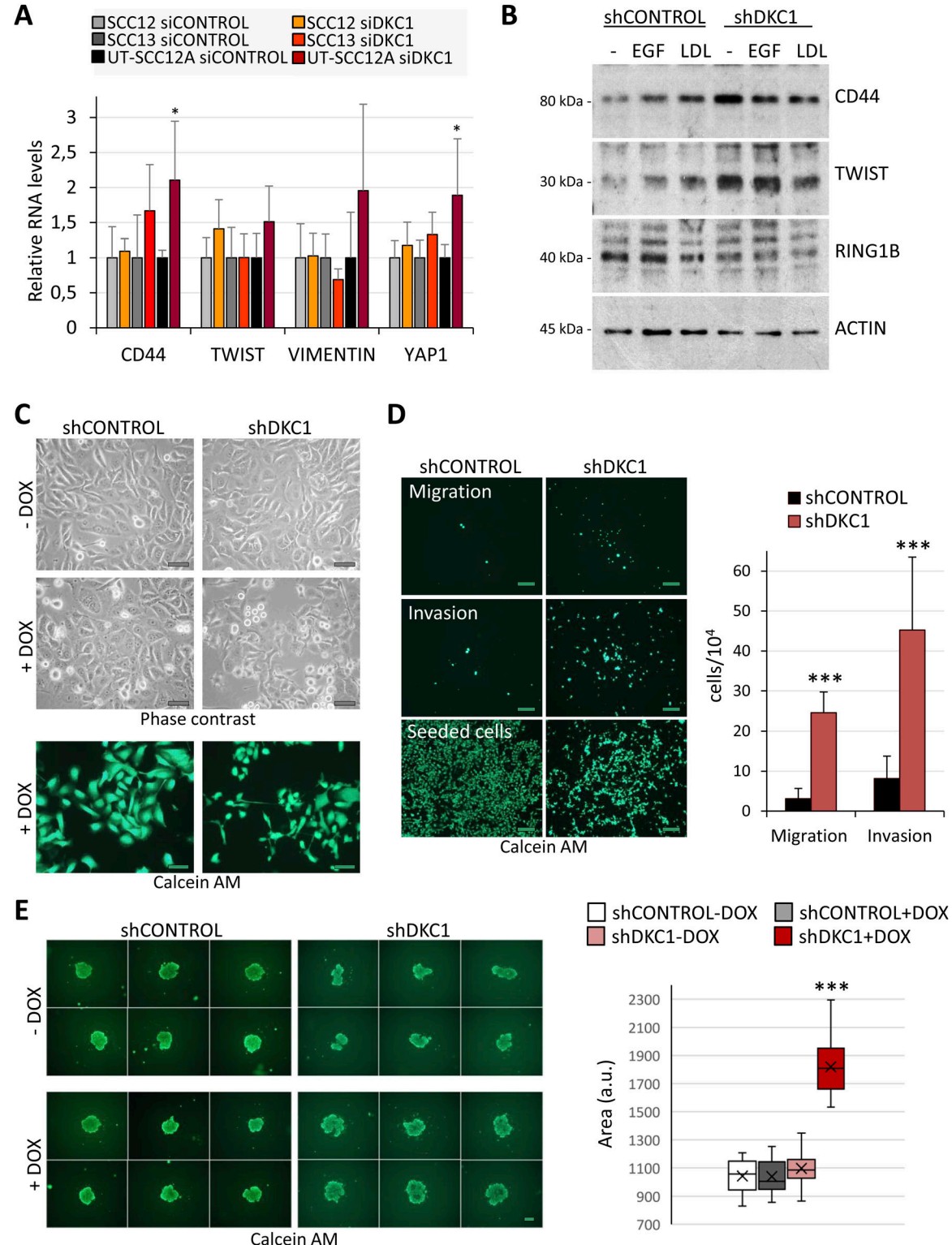

**Figure 3. DKC1-deficient cells acquire metastatic and stemness traits.**
**(A)** Expression of metastasis- and stemness-related genes in cSCC cells with reduced DKC1 levels. Results from two independent experiments performed in triplicate. The t test, *P < 0.05. **(B)** Protein levels of CD44, TWIST, and RING1B in control and DKC1-depleted UT-SCC12A cells in basal conditions or treated with EGF or LDL. ACTIN, loading control. **(C)** Phase-contrast imaging of control and shDKC1 cells, cultured in the absence or presence of doxycycline for 3 d, and the same cultures stained with calcein-AM and visualized by fluorescence microscopy (pictures below). Scale bars, 50 μm. **(D)** In vitro migration and invasion abilities of control and shDKC1 cells cultured with doxycycline for 3 d. Doxycycline-treated cells were seeded into the serum-deprived medium in the top compartments, and migration or invasion through 0.1× Matrigel-coated inserts toward medium supplemented with 10% FBS was quantified 24 h later by calcein-AM staining. Lower pictures, initially seeded cells. Scale bars,

that specifically blocks the mevalonate pathway (Di Bello et al, 2020). Incubation of DKC1-depleted cells with this inhibitor effectively reduced the levels of proteins involved in migration and metastasis (Fig 4C) and impaired their metastatic abilities (Fig 4D), indicating that the changes at the functional level were due to the metabolic rewiring.

### The mevalonate pathway is associated with cSCC progression

Our in vitro experiments demonstrated that DKC1 down-regulation resulted in an enhancement in the mevalonate pathway and the acquisition of metastatic traits in cSCC cells. We therefore wished to investigate whether these findings could be relevant to the clinical progression of cSCC. Because HMGCS1 was one of the most over-expressed genes upon DKC1 depletion, and mevalonate blockade with simvastatin underlies the importance of the HMG-CoA metabolism in migration, we analyzed the expression of this enzyme in cSCC samples. HMGCS1 expression could be detected in the cytoplasm of cells located in the invasive areas of some of these tumors, and the intensity of the staining was strong in few of the tumoral cells (Fig 5A). Correlation analysis of HMGCS1 expression in primary tumors and their pathological features revealed that activation of the mevalonate pathway was associated with tumor size and with the TNM stage (Fig 5B). Importantly, the number of lymphatic metastases expressing HMGCS1 was higher than primary tumors (44% versus 20%, respectively; Fisher's exact test value = 0.012; Fig 5C), supporting the relevance of this pathway for cSCC progression.

In an attempt to identify putative CSCs in cSCC, we also wished to investigate the presence of cells with an enhanced lipid metabolism and CD44 expression in these tumors. Immunohistochemistry revealed that although some tumors display a homogeneous CD44 expression, most cSCCs displayed intratumoral heterogeneity. In these cases, CD44 expression was higher in the invasive areas of the tumor (Fig 5D). Technical constraints regarding CD44 and HMGCS1 compatibility precluded the use of both antibodies to perform immunohistochemistry costaining. To overrule this limitation and identify simultaneous activation of cholesterol biosynthesis/uptake and CD44 expression, we performed co-immunostaining of SREBP2 and CD44. This approach revealed concomitant expression of CD44 and SREBP2 in a subset of cells of the invasive area of the tumor (Fig 5D), suggesting that the metastatic progression might require the acquisition of both metabolic and phenotypic features by the cells of the primary tumors.

## Discussion

Metastasis to the regional lymph node is the most important prognostic indicator for the outcomes of patients with most solid cancers, but the molecular mechanisms involved in cancer cell dissemination from the primary tumor to lymphatics remain obscure. To specifically investigate this initial step, we took advantage of the ability of cSCC, a highly prevalent malignant tumor, to preferentially spread to lymph nodes (Haisma et al, 2016). We have explored the expression of sncRNAs in cSCC samples from patients with non-metastatic and metastatic tumors, and we have observed the reduced expression of a specific subset of sncRNAs, the H/ACA snoRNAs, whereas the number of lymph metastasis expressing DKC1, the core protein with enzymatic activity of the H/ACA RNPs, is lower than DKC1-positive primary tumors. Because the binding of an H/ACA RNA to each RNP stabilizes a single RNA molecule, it is likely that the snoRNA impoverishment observed in metastasizing cSCC results from alterations in RNP formation and maturation because of reduced DKC1 levels, which would in turn affect the expression of a vast number of snoRNAs. This possibility is supported by our own results and by data showing that loss of DKC1 function results in a reduction in the levels of a subset of miRNAs that are encoded within H/ACA snoRNAs (Alawi & Lin, 2011). Possible transcriptional mechanisms accounting for DKC1 down-regulation in cSCC could involve local *DKC1* promoter hypermethylation or mutations/alterations in chromatin remodeling genes (Yilmaz et al, 2017), either working alone or through their interaction with master epithelial transcription factors such as p63 or c-MYC. According to this possibility, we and others have previously reported that poor cSCC differentiation degree associates with an enhanced metastatic risk (Rowe et al, 1992; Toll et al, 2013). In addition, regulation of DKC1 expression could be coordinated, because it has been shown that DKC1 and other protein components of mature H/ACA RNPs, including NHP2 and GAR1, are transcriptionally regulated by c-MYC (Schlosser et al, 2003; Alawi & Lee, 2007; Wu et al, 2008).

DKC1 is a highly evolutionarily conserved protein and the catalytic subunit of the H/ACA RNP. Its enzymatic activity is the RNA pseudouridylation, the most abundant modification in total RNA from human cells, and its main targets are rRNA, sncRNA, snoRNA, and TERC. Reduced pseudouridylation of 28S rRNA leads to dysfunctional translation of key mRNAs, as have been reported for VEGF and p53, whose translation is mediated by internal ribosomal entry site sequences (Rocchi et al, 2013). In addition, defective H/ACA RNP assembly and maturation can affect snoRNA expression and function by canonical-independent mechanisms to regulate mRNA splicing and editing, stress responses, and metabolic homeostasis (Liang et al, 2019). At the functional level, hematopoietic progenitor cells bearing DKC1 mutations associated with bone marrow hypoplasia display reduced differentiation into myeloid and erythroid cells (Bellodi et al, 2013), and DKC1 and the H/ACA RNP are also involved in the self-renewal and differentiation of mesenchymal stem cells (Zhang et al, 2017).

Inactivating mutations of DKC1 cause dyskeratosis congenita, a rare genetic condition that results in bone marrow failure and a predisposition for cancer, frequently SCC, mainly caused by a lack of pseudouridylation on rRNA (Ruggero et al, 2003). The role of DKC1 as a tumor suppressor protein, likely through dysfunctional translation, is supported by the clinical features of patients bearing DKC1

200 $\mu$m. On the right, data obtained from two independent experiments performed in triplicate. Statistical analysis was performed by the *t* test, \*\*\**P* < 0.001. **(E)** Images from one representative experiment of two showing spheroids formed 7 d after plating shControl and shDKC1 UT-SCC12A cells with vehicle (−DOX) or with doxycycline (+DOX) (left pictures) (scale bars, 200 $\mu$m). On the right, graph showing the quantification of the spheroid areas (a.u.) (left). Mean ± s.e.m., two-tailed *t* test.

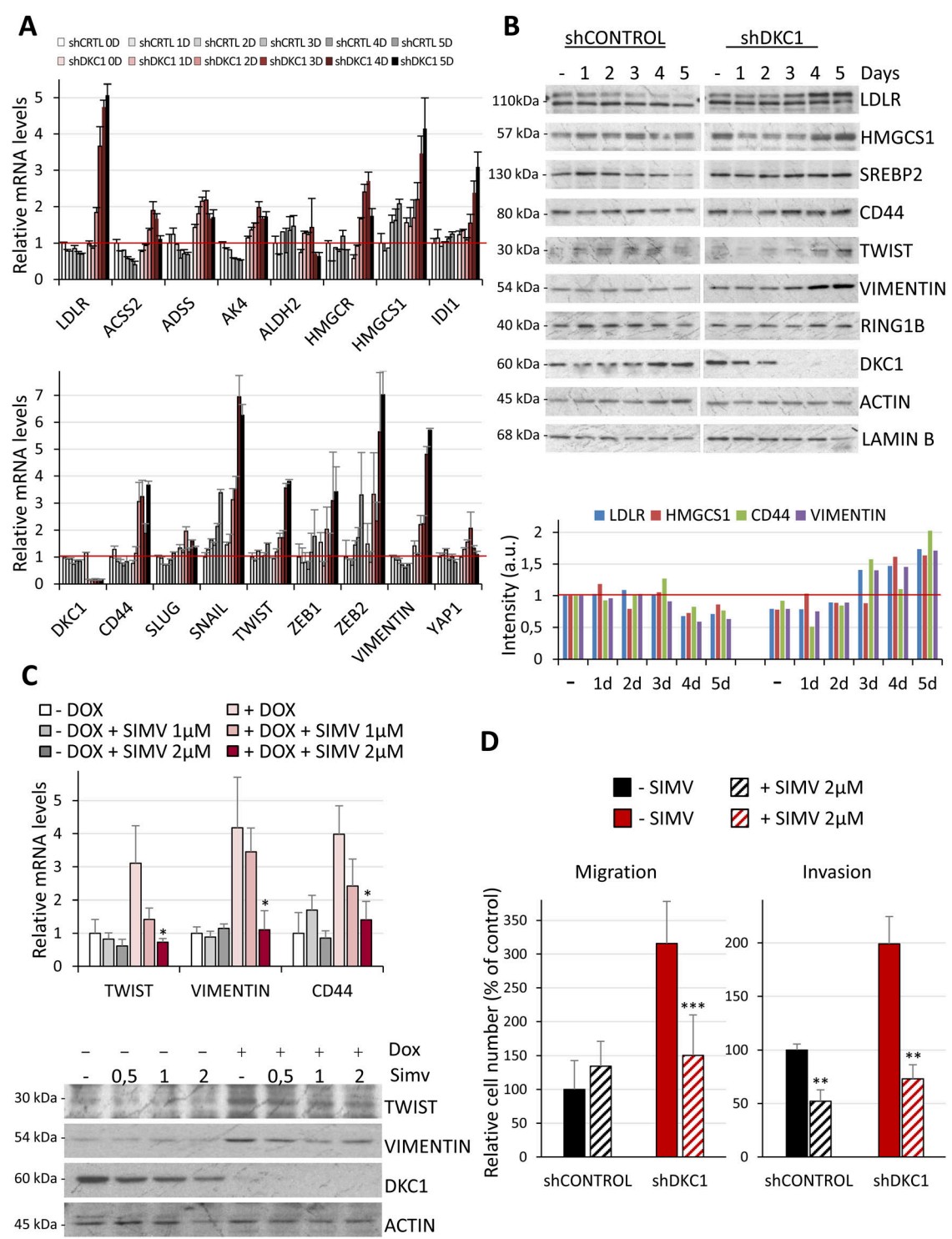

**Figure 4. DKC1 depletion promotes transcriptional changes in lipid metabolism and metastatic genes.**
**(A)** Time-course analysis of the expression of lipid metabolism (upper panel), and metastatic genes (lower panel) in control and DKC1-depleted cells, determined by qRT–PCR. **(B)** Time-course analysis of lipid, mesenchymal, and DKC1 proteins in control and DKC1-depleted cells, determined by Western blot. LAMIN B was used as a loading control. Lower panel, densitometric quantification of Western blot signals, normalized to LAMIN B intensity. **(C)** Effect of simvastatin in the expression of proteins involved in migration and invasion. UT-SCC12A carrying inducible shDKC1 cultured in the absence or presence of doxycycline was treated with different concentrations of simvastatin for 48 h. Figure shows the results from one representative of two independent experiments performed in triplicate, *$P < 0.05$. **(D)** Effect of simvastatin on the in vitro migration and invasion abilities of control and shDKC1 cells cultured with doxycycline, as indicated in Fig 3D. Data were obtained from two independent experiments performed in triplicate. Statistical analysis was performed by the $t$ test, ***$P < 0.001$ and **$P < 0.005$.

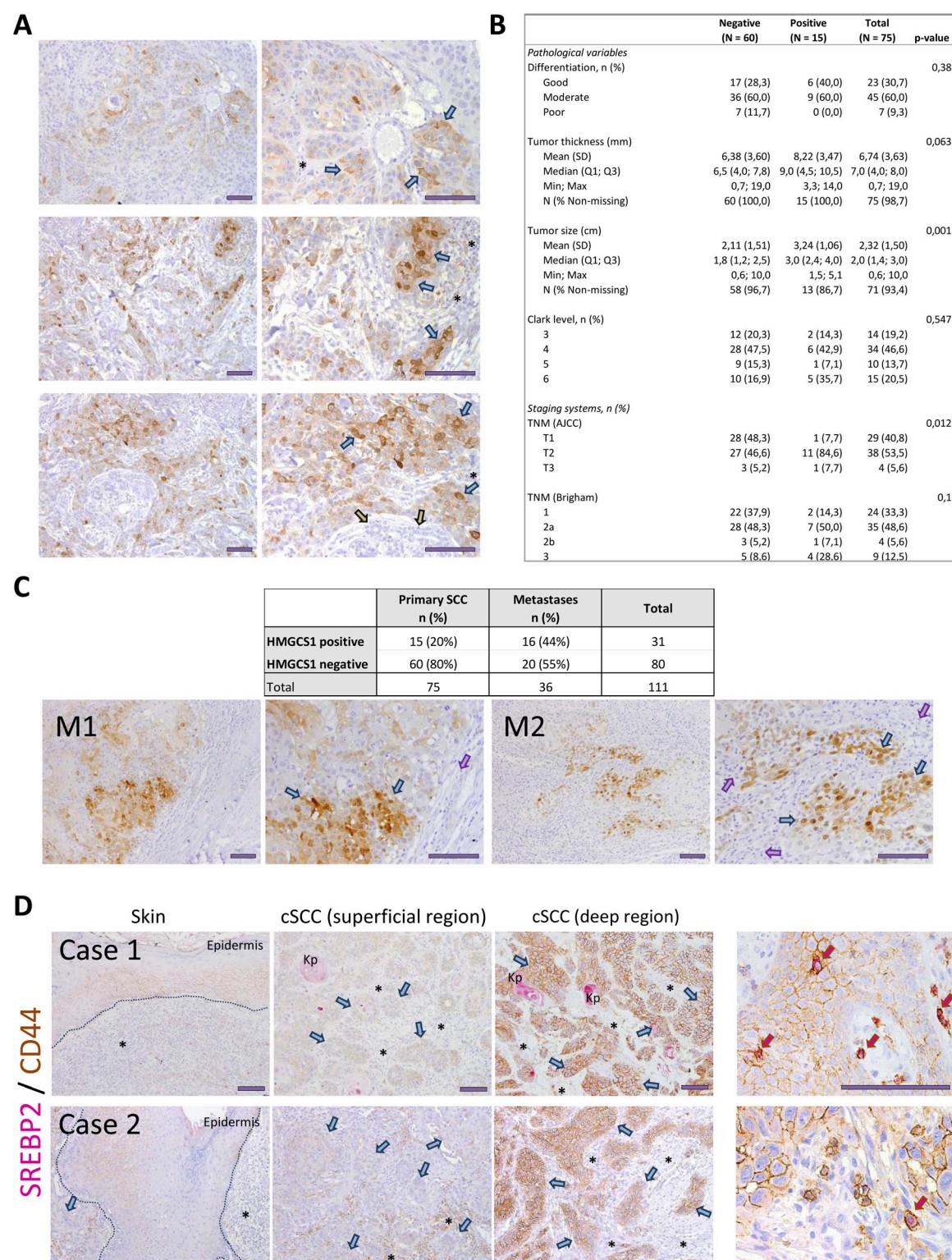

**Figure 5. Expression of lipid metabolism markers in cSCC samples.**
**(A)** HMGCS1 expression in three primary cSCC samples (two different magnifications), detected by immunohistochemistry. The intense cytoplasmic staining of some tumoral cells was observed (blue arrows). Gray arrows, eccrine glands; asterisks, inflammatory infiltrate. **(B)** Correlation analysis of HMGCS1 expression and pathological features of the samples. Statistical analyses were performed with the Mann–Whitney U test and Fisher's exact test for continuous and categorical variables, respectively. **(C)** Contingency table of HMGCS1 expression in primary cSCC and metastases. Pictures below, HMGCS1 expression in two representative lymph metastases at two different magnifications, detected by immunohistochemistry. Tumoral cells, blue arrows; lymphoid tissue, purple arrows. **(D)** Expression of CD44 (brown, membrane) and

loss-of-function alterations and by experimental murine models. Paradoxically, DKC1 overexpression correlates with disease progression and poor prognosis in many sporadic cancers, such as prostate and breast tumors, and DKC1 levels also associate with higher levels of TERC expression and telomerase activity in some of these tumors. These apparently contradictory functions could reflect cell-specific balances of different DKC1 functions in each tumor type. We have investigated the functional effects of DKC1 down-regulation in cSCC cells and unveiled new roles for DKC1 in the metastatic process: metabolism and invasiveness. Transcriptional, proteomic, and metabolomic analyses showed that reduced DKC1 levels in cSCC cells resulted in both an activation of the mevalonate pathway and the acquisition of an invasive phenotype. Rewiring the cholesterol metabolism via the mevalonate pathway allows the tumoral cell to migrate, to adapt, and to survive in the rapidly changing conditions of the tumoral microenvironment, and must be of paramount relevance for those tumoral cells that escape from the primary tumor through a lymphatic vessel. Our results showing HMGCS1 expression in cells of the primary tumor would indicate that the metastatic cell has already adapted its metabolism and is prone to invade and to benefit from the high concentrations of cholesterol and other lipids in the lymphatics, which are responsible for the drainage of their excessive accumulation from peripheral tissues (Huang et al, 2015). Indeed, the up-regulation of enzymes and proteins that regulate an important part of the lipid metabolism allows energy supply, changes membrane composition and permeability, and provides molecular substrates for the epigenetic regulation of the gene expression (Fu et al, 2021). Our integrated analysis of gene expression in DKC1-depleted cells pointed to the acetyl-CoA/mevalonate pathway as the central node for the metabolism in these cells, which would be fueled by fatty acid uptake through LDLR and by TCA metabolites. This metabolic switch would directly impact on the migratory and invasive abilities of cSCC cells, specifically required in the most early metastatic step, in agreement with data reporting the association of acetyl-CoA hydrolysis with hepatocellular carcinoma (HCC) metastasis through the epigenetic regulation of *TWIST2* (Lu et al, 2019), and could also explain the changes observed in cytoskeleton remodeling and the enhanced endocytic Ras-related protein Rab-5A/Rab11 trafficking in breast cancer and osteosarcoma cells with reduced DKC1 levels (Di Maio et al, 2017). Sustaining this possibility, the mevalonate pathway has been reported to regulate cell size homeostasis and proteostasis through geranylgeranylation of the small GTPase RAB11 (Miettinen & Björklund, 2015). Our results in metastatic cSCC parallel with those that associate the expression of the fatty acid receptor CD36 with metastasis in oral SCC metastasis-initiating cells and in experimental metastasis models of human melanoma and breast cancer (Pascual et al, 2018), and underscore the notion that the enhanced expression of specific fatty acid receptors in each tumor type is a key feature for the acquisition of metastatic traits. Although patients with tumors relying exclusively on the exogenous support of fatty acids for the metastatic progression could potentially benefit from a dietary intervention, in cSCC a

therapeutic intervention aimed to block essential genes for controlling cholesterol metabolism will be required. Drugs targeting some of these genes are under preclinical investigation, including statins that target HMGCR, a rate-limiting enzyme in cholesterol metabolism (Fu et al, 2021). These drugs would be effective before tumor spreading, because changes induced upon DKC1 down-regulation are very likely transient and restricted to the initial metastatic step to the lymphatics, whereas tumoral cells growing in the primary tumor and in metastases use glucose as the main energy source, property that constitutes the base for the detection and staging of cancer by positron emission tomography with the glucose analogue tracer 2-[fluorine-18]fluoro-2-deoxy-d-glucose (FDG).

In summary, loss of DKC1 function results in a rewiring of the metabolism and the acquisition of invasive features in cSCC, and likely in other tumors of epithelial origin. The precise mechanisms by which DKC1 is down-regulated, and the direct targets affected by the loss of DKC1 function and the associated snoRNAs remain unknown, and further studies will be required to address the tumor suppressor role of the DKC1 complex in cancer and in syndromes associated with H/ACA RNP alterations. Overall, our results show the importance of the mevalonate pathway for the metastatic spread, and they will allow new therapeutic strategies taking advantage of this potential vulnerability of the cancer cell.

# Materials and Methods

### Patients

Six tertiary care hospitals of Spain participated in this study, and the approval to conduct this study was obtained from the ethics committees from all the centers in accordance with the guidelines of the Helsinki Declaration of 1975, as revised in 1983. Patients who had developed MSCCs between 2001 and 2011 were retrieved from local skin cancer registries. 56 primary metastasizing cSCCs that had evolved to histologically confirmed lymph node (n = 51) or cutaneous in-transit (n = 5) metastases and a control group of 51 patients with cSCCs who had not developed any metastasis in a 5-yr follow-up period were included in the study. 39 metastases (35 lymph node/four cutaneous in-transit) from 39 patients were also evaluated. Histopathological evaluation of the samples has already been described (Toll et al, 2013).

### Immunohistochemical procedures

These protocols were performed following the standard techniques. Briefly, immunohistochemistry was performed on 4-mm-thick sections. Antigen retrieval was performed in 10 mM citrate (pH 6) for 15 min in a pressure cooker. The slides were then incubated with primary antibodies for 12 h. After washing, the EnVision+ System/HRP antibody reagent was applied (Dako, Agilent

---

SREBP2 (pink, cytoplasm) in four different fields of two primary cSCC tumors. The basal layer of normal epidermis is outlined with a dashed line. Tumoral cells, blue arrows; asterisks, inflammatory infiltrate; Kp, keratin pearls. Pictures on the right, high magnification of fields in the same cases. The presence of scarce cells with a concomitant expression of high SREBP2 and CD44 levels was observed. Scale bars, 100 $\mu m$.

Technologies). Immunohistochemical reactions were developed using DAB as a chromogenic substrate. Sections were counterstained with hematoxylin, dehydrated, and mounted. For sequential HRP/AP immunoenzymatic double staining analysis of CD44 and SREBP2, sections were first stained with primary anti-CD44, followed by secondary HRP-conjugated antibody. Subsequently, slides were washed extensively and incubated with anti-SREBP2 primary antibody followed by incubation with AP-conjugated secondary antibody. Finally, HRP activity was developed as above, and AP detection was performed by incubation with K1395 Fast Red reagent (Dako). Sections were counterstained with hematoxylin, dehydrated, and mounted.

Immunohistochemical staining was scored by two independent observers. Evaluation of DKC1 and HMGCS1 was performed to classify samples as negative or positive.

## Cell culture

SCC13 and SCC12 cells were a generous gift from Dr. S Aznar-Benitah (IRB). UT-SCC12A cells were a generous gift from Dr. R Grénman (University of Turku). H520 cells were a generous gift from Dr. J Alcaraz (University of Barcelona). Transient expression of short interfering RNAs (siRNA; Dharmacon) was performed by oligofection as described earlier (Hernández-Ruiz et al, 2018). Sequences for siRNA and shRNA (short hairpin RNA) can be found in Table 1.

shRNAs were cloned into pTET-LKO-Puromycin vector (plasmid #21915; Addgene). To generate UT-SCC12A stably expressing inducible shRNAs, retroviral supernatants from Phoenix cells transfected with shRNA vectors were used to infect the cells, which were selected with 2 μg/ml puromycin (Sigma-Aldrich). shRNA expression was induced with 100 ng/ml doxycycline in cell culture in DMEM supplemented with 10% tetracycline-free FBS. When indicated, cells were cultured with EGF (50 ng/ml) and LDL (100 μg/ml) (both from PeproTech), or simvastatin (Sigma-Aldrich) in Earle's salt medium.

## RNA isolation, qRT–PCR, and expression arrays

RNA from cSCC was purified with miRNeasy FFPE Kit (QIAGEN) and processed to carry out the miRNA 4.0 (Applied Biosystems) array, following the FlashTag Biotin HSR RNA Labeling Kit (P/N 703095, Rev.2) (Applied Biosystems) and Expression Wash, Stain and Scan User Manual (P/N 702731 Rev. 3) (Applied Biosystems). For sncRNA expression arrays, samples were amplified and labeled according to the GeneChip manufacturing protocol. For the statistical analysis, R programming (version 4.0.3) was used, together with different packages from Bioconductor (Gentleman et al, 2004) and the Comprehensive R Archive Network (CRAN 2019). After quality control of raw data, samples were background-corrected, quantile-normalized, and summarized to a gene level using the robust multi-chip average (RMA) (Irizarry et al, 2003). An empirical Bayes moderated t-statistics model (LIMMA) (Ritchie et al, 2015) was built to detect differentially expressed genes between the studied conditions.

RNA isolation from cell cultures and qRT–PCR were performed following the standard techniques. RNA was isolated with GenElute Total Mammalian RNA Kit (Sigma-Aldrich), and cDNA was obtained using Transcriptor First Strand cDNA Synthesis Kit (Roche). For gene expression arrays, samples were processed at MARGenomics IMIM's core facility following the user's manuals (Ambion WT Expression PN 4425209, Revised B 05/2009, Thermo Fisher Scientific, Applied Biosystems; WT Terminal Labeling and Hybridization PN 702808; and Expression, Wash, Stain and Scan PN 702731, Applied Biosystems). The expression platform used was Clariom S Human microarray (Thermo Fisher Scientific, Applied Biosystems). Correction for multiple comparisons was performed using false discovery rate (Benjamini & Hochberg, 1995), and adjusted P-values were obtained. The ranked list of genes was generated using the –log(P-val)*signFC for each gene from the statistics obtained in the DE analysis with LIMMA (Ritchie et al, 2015).

qRT-PCRs were performed in triplicate in 384-well plates and were run in an ABI PRISM 7900HT Real-Time PCR cycler. qRT–PCR to analyze gene expression was performed using SYBR Green PCR Master Mix (Applied Biosystems, Life Technologies), and TBP expression was used as endogenous control for normalization purposes. Primers used for qRT–PCR can be found in Table 2.

## Proteomic analysis

The proteomic analysis was performed in triplicate samples of protein extracts from UT-SCC12A cells transiently transfected with control or DKC1 siRNA. Samples (10 μg) were precipitated with 6 vol of cold acetone, and the pellet was dissolved in 10 μl of 6 M urea/ 200 mM ammonium bicarbonate. After precipitation and dissolution, samples were reduced with dithiothreitol (30 nmol, 37°C, 60 min) and alkylated in the dark with iodoacetamide (60 nmol, 25°C, 30 min). The resulting protein extract was first diluted to 2 M urea with 200 mM ammonium bicarbonate for digestion with endo-proteinase LysC (1:10 w:w, 37°C, o/n; Wako), and then diluted twofold with 200 mM ammonium bicarbonate for trypsin digestion (1:10 w:w, 37°C, 8 h; Promega). After digestion, peptide mix was acidified with formic acid and desalted with a MicroSpin C18 column (The Nest Group, Inc.) before LC-MS/MS analysis. Samples were analyzed using an Orbitrap Fusion Lumos mass spectrometer (Thermo Fisher Scientific) coupled to an EASY-nLC 1200 (Thermo Fisher Scientific [Proxeon]). Peptides were loaded directly onto the analytical column and were separated by reversed-phase chromatography using a 50-cm column with an inner diameter of 75 μm, packed with a 2-μm C18 particles spectrometer (Thermo Fisher Scientific). Chromatographic gradients started at 95% buffer A and 5% buffer B with a flow rate of 300 nl/min and gradually increased to 25% buffer B and 75% A in 79 min and then to 40% buffer B and 60% A in 11 min. After each analysis, the column was washed for 10 min with 100% buffer B (buffer A: 0.1% formic acid in water; buffer B: 0.1% formic acid in 80% acetonitrile). The mass spectrometer was operated in a positive ionization mode with nanospray voltage set at 2.4 kV and source temperature at 275°C. The acquisition was performed in a data-dependent acquisition mode, and full MS scans with 1 microscans at a resolution of 120,000 were used over a mass range of m/z 350–1,500 with detection in the Orbitrap mass analyzer. Automatic gain control was set to 1E5 and injection time to 50 ms. In each cycle of data-dependent acquisition analysis, after each survey scan, the most intense ions above a threshold ion count of 10,000 were selected for fragmentation. The number of selected

## Life Science Alliance

**Table 1.  Sequences of siRNA and shRNA.**

| siRNA | Sequence | Brand |
|---|---|---|
| DKC1HSS102781 | GGCCAAGAUUAUGCUUCCAGGUGUU | Invitrogen |
| DKC1HSS102782 | GGAUCCCAAGGUGACUGGUUGUUUA | Invitrogen |
| DKC1HSS176185 | GAAGUCACAACAGAGUGCAGGCAAA | Invitrogen |
| shRNA | Sequence | |
| shControl | GCTGACCCTGAAGTTCATC | |
| shDKC1 | GCTCAGTGAAATGCTGTAGAA | |

precursor ions for fragmentation was determined by the "Top Speed" acquisition algorithm and a dynamic exclusion of 60 s. Fragment ion spectra were produced via high-energy collision dissociation (HCD) at normalized collision energy of 28%, and they were acquired in the ion trap mass analyzer. Automatic gain control was set to 1E4, and an isolation window of 1.6 m/z and a maximum injection time of 200 ms were used. Digested bovine serum albumin (New England Biolabs) was analyzed between each sample to avoid sample carryover and to assure stability of the instrument, and QCloud (Chiva et al, 2018) has been used to control instrument longitudinal performance during the project. Acquired spectra were analyzed using the Proteome Discoverer software suite (v2.0; Thermo Fisher Scientific) and the Mascot search engine (v2.6; Matrix Science [Perkins et al, 1999]). The data were searched against a Swiss-Prot human database (as in October 2018, 20408 entries) plus a list (Beer et al, 2017) of common contaminants and all the corresponding decoy entries. For peptide identification, a precursor ion mass tolerance of 7 ppm was used for MS1 level, trypsin was chosen as enzyme, and up to three missed cleavages were allowed. The fragment ion mass tolerance was set to 0.5 D for MS2 spectra. Oxidation of methionine and N-terminal protein acetylation were used as variable modifications, whereas carbamidomethylation on

**Table 2.  Sequences of primers for qRT-PCR.**

| SNORA10 | CTCAGCTCCGCTTAACCACA | TGTCGTGCATTAGGAGAGCC |
|---|---|---|
| SNORA18 | TAGCCTGCACATCGTTGGAA | TCCCACAGATCTAACAATAGTCCT |
| SNORA40 | AACGTGGACAAAGACTTACAGA | TGAACAATGAGTTCTGGGTTGC |
| SNORA64 | TCTCGGCTCTGCATAGTTGC | CATCCGAGGTCCTGAGGAGA |
| DKC1 | CAGTAAATGCCATCTGCTATG | ACAATCTCCTGATTGACCTC |
| ACSS2 | GCTCAAGAAGCAGATTAGAG | CATGGTCATTCTGAGCAATC |
| ACAT1 | CTCGATGCAATAATCCTTTCTC | TCCAATTGGGATGTCTGG |
| ADSS | AGATGAGCTTCAAATTCCAG | GTAGGGCTTCAAACTTACTTC |
| AK4 | ATTATACAAGAGCCGAGGAG | GAGAAAAGTGTGTAAACGTAGG |
| ALDH2 | ACACTGAAGTGAAAACTGTC | ATCTTGCTGAACTTTCCATC |
| BCAT2 | GGGAACCATGAACATCTTTG | CATGTCCAGTAGACTCTGTC |
| HADHB | CCTTCAAAGTACCAGGAAAAG | CACCATCAGTCAAGAAAGAAG |
| HMGCS1 | TTGGCTTCATGATCTTTCAC | AATTTAACATCCCCAAAGGC |
| HMGCR | TGAGGGCTCCTTCCGCTCCG | ACTAGAGGCCACCGAACCCCG |
| IDI1 | AATAAACACTAACCACCTCG | CTCGATGCAATAATCCTTTCTC |
| MCCC2 | AGAAAGTCTGGAGTAAGTGAC | CTCTTCAGAAGGTTCAATGG |
| NAPRT1 | GTGAGGTGAATGTCATTGG | GGCCACCAGCTTATAGAC |
| PSAT1 | GAGTTTGACTTTATACCCGATG | ATCACACCAAACTTGGAAAC |
| SHMT2 | CATTTGAGGACCGAATCAAC | CACCTGATACCAGTGAGTAG |
| CD44 | TGGCACCCGCTATGTCGA | GTAGCAGGGATTCTGTCT |
| SLUG | CAGGGAACTGGACACACATACA | AGGATCTCTGGTTGTGGTATGACA |
| SNAIL | ACCCCAATCGGAAGCCTAAC | TGGTCGTAGGGCYGCTGGAA |
| TWIST | CCGGAGACCTAGATGTCATTGTT | TTTTAGTTATCCAGCTCCAGAGTCTCT |
| ZEB1 | GTTACCAGGGAGGAGCAGTG | TCTTGCCCTTCCTTTCTGTCA |
| ZEB2 | TGCAAGAGGCGCAAACAAG | AGAACCTGTGTCCACTACATTGTCA |
| VIMENTIN | CCTCCGGGAGAAATTGCA | GCATTGTCAACATCCTGTCTGAA |
| YAP1 | CGCTCTTCAACGCCGTCA | AGTACTGGCCTGTCGGGAGT |
| LDLR | GAGGACAAAGTATTTTGGACAG | GTAGGTTTTCAGCCAACAAG |
| GAPDH | CTTCAACAGCGACACCCACT | GTGGTCCAGGGGTCTTACTC |
| TBP | TTCGGAGAGTTCTGGGATTGTA | TGGACTGTTCTTCACTCTTGGC |

**Table 3. List of antibodies.**

| Antibody | Catalog number | Brand | Application |
|---|---|---|---|
| DKC1 | HPA001022 | Sigma-Aldrich | IHC, WB |
| HMGCS1 | SC-166763 | Santa Cruz | IHC, WB |
| SREBP2 | SC-271615 | Santa Cruz | IHC, WB |
| CD44 | 636498500 | Roche | IHC |
| ACSS2 | HPA004141 | Sigma-Aldrich | WB |
| LDLR | ZRB1176 | Sigma-Aldrich | WB |
| IGF1R | 9750 | Cell Signaling | WB |
| EZH2 | 5246 | Cell Signaling | WB |
| S2P | SC-293341 | Santa Cruz | WB |
| CD44 | 103004 | BioLegend | WB |
| TWIST | ab50887 | Abcam | WB |
| RING1B | d139-3 | Millipore | WB |
| VIMENTIN | M0725 | Dako | WB |
| LAMIN B1 | ab16048 | Abcam | WB |
| ACTIN | A2228 | Sigma-Aldrich | WB |
| HRP-conjugated rabbit antibody | P0260 | Dako | WB |
| HRP-conjugated mouse antibody | P0448 | Dako | WB |

cysteines was set as a fixed modification. False discovery rate in peptide identification was set to a maximum of 5%. Peptide quantification data were retrieved from the "Precursor ion area detector" node from Proteome Discoverer (v2.0) using 2 ppm mass tolerance for the peptide extracted ion current (XIC). The sum of the three most intense peptides per proteins was used to estimate protein abundance. Estimated protein abundance was log-transformed, normalized using a median normalization, prior statistical inference. Fold change, $t$ test, and adjusted $P$-value (q-value) were calculated to compare controls with siDKC1-expressing cells.

### Targeted determination of TCA metabolites

Acidic metabolites belonging to TCA and related pathways (ketone bodies and glycolysis) were determined based on a previously reported method (Gomez-Gomez et al, 2022). Briefly, after addition of the internal standard (consisting in a mixture of labeled analytes), 25 μl of the lysate extract was derivatized with o-benzyl-hydroxylamine. The reaction was stopped by addition of 1 ml of water, and analytes were extracted with ethyl acetate. After evaporation of the organic layer, the extract was reconstituted into 150 μl of water. 10 μl of the reconstituted extract was injected in the LC-MS/MS system consisting of an Acquity I-Class UPLC system coupled to a triple quadrupole (TQS Micro) mass spectrometer (Waters Associates). The separation was achieved at 55$^0$C using an Acquity BEH C18 column (100 × 2.1 mm i.d., 1.7 $\mu$m) (Waters Associates) at a flow rate of 300 $\mu$l/min. Water–ammonium formate (1 mM)/formic acid (0.01%) and methanol–ammonium formate (1 mM)/formic acid (0.01%) were used as mobile phases.

Analytes were determined by selected reaction monitoring using the most specific transition for their determination. MassLynx software v4.1 and TargetLynx XS were used for data management.

### Protein extraction and Western blot

These protocols were performed following the standard techniques. The list of antibodies used can be found in Table 3.

Western blot quantification was performed using ImageJ densitometry software. Intensity of individual bands was normalized to ACTIN signal and referred to control conditions.

### LDL uptake assay

LDL uptake was analyzed using a BODIPY-R FL kit (Thermo Fisher Scientific), following the manufacturer's instructions. Briefly, UT-SCC12A cells carrying inducible control or DKC1 shRNA and cultured in the presence of doxycycline were seeded into 48-well plates at 4 × 10$^4$ cells/well. After serum deprivation for 16 h, cells were pretreated without or with unlabeled LDL for 30 min at 37°C. Then, cells were incubated with BODIPY TM FL LDL (5 $\mu$g/ml) for 3 h, washed with PBS, and fixed with isopropyl alcohol. LDL deposits were quantified by analyzing more than 10 different fields per condition using ImageJ software.

### Migration, invasion, and spheroid formation

Migration and invasion experiments were performed with UT-SCC12A cells carrying inducible control or DKC1 shRNA and cultured in the

presence of doxycycline, following the standard protocols. Briefly, cells were washed with serum-free medium, resuspended in serum-free medium, and seeded in the top compartments of 8 $\mu$M transwells containing DMEM supplemented with 10% tetracycline-free FBS in the bottom chambers. Cell invasion assays were performed as above, and 0.1× basement extracellular membrane (BME) Matrigel was used in each well to coat the top chamber of the 96-well invasion device.

Spheroid formation ability was checked by seeding 4,000 cells on ultra-low adherent plates, and analyzed 7 d after plating. Cell staining was performed with 1 M calcein-AM (Sigma-Aldrich).

# Data Availability

Data from array analyses have been deposited in GEO, accession GSE207545 (https://www.ncbi.nlm.nih.gov/geo/query/acc.cgi?acc=GSE207545). The mass spectrometry proteomic data have been deposited at the ProteomeXchange Consortium via the PRIDE (Vizcaíno et al, 2016) repository with identifier PXD034155.

# Supplementary Information

# Acknowledgements

The authors would like to thank Lara Nonell and Xavier Duran for their mathematics support, and Júlia Perera and Ariadna Acedo for their help with GEO accessions. This study was supported by the grant PI18/00065 funded by Instituto de Salud Carlos III (ISCIII) and cofunded by the European Union, Ministerio de Economía y Competitividad, Spain, and from the "Xarxa de Bancs de tumors sponsored by Pla Director d'Oncologia de Catalunya (XBTC)." The CRG/UPF Proteomics Unit is part of the Spanish Infrastructure for Omics Technologies (ICTS OmicsTech), and it is supported by "Secretaria d'Universitats i Recerca del Departament d'Economia i Coneixement de la Generalitat de Catalunya" (2017SGR595). We also acknowledge support of the Spanish Ministry of Science and Innovation to the EMBL Partnership, the Centro de Excelencia Severo Ochoa, and the "CERCA Programme/Generalitat de Catalunya."

## Author Contributions

E Andrades: conceptualization, data curation, validation, methodology, and writing—original draft.
A Toll: conceptualization, resources, funding acquisition, investigation, data curation, methodology, and writing—original draft.
G Deza: resources, data curation, investigation, methodology, and writing—original draft.
S Segura: resources, data curation, investigation, methodology, and writing—original draft.
R Gimeno: conceptualization, investigation, methodology, and writing—original draft, review, and editing.
G Espadas: data curation, software, formal analysis, methodology, and writing—original draft.
E Sabidó: data curation, software, formal analysis, methodology, and writing—original draft.
N Haro: data curation, software, formal analysis, methodology, and writing—original draft.
ÓJ Pozo: data curation, software, formal analysis, methodology, and writing—original draft.
M Bódalo: data curation, software, formal analysis, methodology, and writing—original draft.
P Torres: investigation, methodology, and writing—original draft.
RM Pujol: conceptualization, resources, supervision, funding acquisition, validation, investigation, visualization, methodology, and writing—original draft, review, and editing.
I Hernández-Muñoz: conceptualization, resources, data curation, formal analysis, supervision, funding acquisition, validation, investigation, visualization, methodology, project administration, and writing—original draft, review, and editing.

## Conflict of Interest Statement

The authors declare that they have no conflict of interest.

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
