## [Reviewer comments · Life Science Alliance]

Life Science Alliance

LOSS OF DYSKERIN FACILITATES THE ACQUISITION OF METASTATIC TRAITS BY ALTERING THE MEVALONATE PATHWAY

Evelyn Andrades, Agusti Toll, Gustavo Deza, Sonia Segura, Ramón Gimeno, Guadalupe Espadas, Eduard Sabido, Noemí Haro, Oscar Pozo, Marta Bódalo, Paloma Torres, Ramon Pujol, and Inma Hernandez-Munoz

DOI: <https://doi.org/10.26508/lsa.202201692>

Corresponding author(s): *Inma Hernandez-Munoz, Institut Hospital del Mar d'Investigacions Mèdiques*

Review Timeline:

Submission Date:	2022-08-26
Editorial Decision:	2022-10-21
Revision Received:	2022-12-22
Editorial Decision:	2023-01-12
Revision Received:	2023-01-19
Accepted:	2023-01-19

Transaction Report:

October 21, 2022

Re: Life Science Alliance manuscript #LSA-2022-01692-T

Dr. Inma Hernandez-Munoz
IMIM-Hospital del Mar
Cancer Research Program
Doctor Aiguader, 88
Barcelona 8003
Spain

Dear Dr. Hernandez-Munoz,

Thank you for submitting your manuscript entitled "LOSS OF DYSKERIN FACILITATES THE ACQUISITION OF METASTATIC TRAITS BY ALTERING CHOLESTEROL METABOLISM VIA THE MEVALONATE PATHWAY" to Life Science Alliance. The manuscript was assessed by expert reviewers, whose comments are appended to this letter. We invite you to submit a revised manuscript addressing the Reviewer comments.

Thank you for this interesting contribution to Life Science Alliance. We are looking forward to receiving your revised manuscript.

Sincerely,

B. MANUSCRIPT ORGANIZATION AND FORMATTING:

Reviewer #1 (Comments to the Authors (Required)):

In this research report, Andrades and coworkers studied the expression of small non-coding RNAs in cutaneous cell carcinoma (cSCC). They have identified a global reduction in the expression of small nucleolar RNAs (snoRNAs) in primary cSCC associated with metastasis. Reasoning that such a global effect might be due to post-transcriptional mechanisms, the authors investigated the expression of Dyskerin (DKC1), a protein regulating snoRNA stability. Having established that down regulation of DKC1 reduced H/ACA snoRNA levels, the authors investigated DKC1 expression in cSCC primary tumors and metastases and concluded that decreased expression of DKC1 is associated with cSCC metastatic progression. Then the authors studied the functional impact of DKC1 in UT-SCC12A cells, by transcriptomic and proteomic approaches and identified 82 upregulated mRNA/proteins among which a number corresponded to enzymes and proteins involved in cholesterol metabolism. They also found a correlation between reduction of DKC1 and acquisition of an EMT phenotype and stemness properties. Next authors interrogated the link between cholesterol homeostasis and metastasis in cSCC. Using a pharmacological approach, they showed that statin treatment abrogated the induction of Twist and Vimentin (two EMT genes) in DKC1 depleted cells. Next to have a clinical relevance for their finding made a correlation analysis for HMGS1 expression by immunohistochemistry and pathological features in a series of cSCC cases and claim an association with lymph node metastasis and CD44 expression.

Altogether this work shows new insights into the role of small non coding RNAs in cancer. They show some convincing data on the relationship between DKC1 expression, lipid metabolism and cancer progression. However, some claims are not fully justified and further clarification are needed.

Here are my concerns:

The additional files should be better organized in figure and table, in the text calling additional files instead of supplementary figure is confusing.

1- Page 3 line 9 DKC1 is not defined, line 14 HMGS1 is not defined same in page 6 line 9

2- Material and methods, page 7 IHC the scoring method is not explained, did the author use a threshold for positivity?

3- Results, page 12 lines 4-7, the authors should mention the cell line used in the text and in figure 1D and comment their choice. If I am correct, in Fig. 1D, the authors use SCC13 to show the link between reduction of DKC1 and snoRNAs expression. This is also substantiated in supplementary Figure 1 with SCC12 and UT-SCC12A cells. I understood that SCC12 and 13 were differentiated cSCC cell line, not representative of Metastatic MSCC. However, the effect of siDKC1 is similar than in the metastatic UT-SCC12A. Is there an explanation?

4- Page 11, the author did a validation in an independent cohort which reinforces their finding, could they comment on the selection of SnoRNA18?

5- A clinical study on DKC1 expression was performed on tissue microarrays page 12 lines 7-13, I did not find the data leading to the conclusions. Figure 5E is an illustration of the microarray series I suppose? It should indicate that DKC1 is labelled, is there an H-score? The subpanels should be indicated and are Metastases from the same patient than MSCC? The scale bar is difficult to see.

6- Regarding gene upregulated in the absence of DCK1, page 13, lines 13 it is stated that 82 were found. Then the meaning of lines 14-15 is very unclear, "42 beard catalytic activity and 45 were involved in acetylation processes". Beard means bore? had? Figure 2 A is difficult to understand the red and dots legend could be added directly on the figure. Line 19 ACSS2 is not defined. Figure 2G lacks size bars and is there quantification?

7- Regarding migration and invasion effects, Figure 3, C, D and E lacks subpanel names and size bars and it seems that less cells were seeded in shDKC1? On page 19, lines 3-4, the authors mention a link between simvastatin and cell migration but there are no data on cell migration and simvastatin treatment. However, on Figure 4C simvastatin seems to reduce DKC1 expression in cells not treated with Doxycycline.

8- On page 19 and on Figure 5, the authors studied the relevance of their finding in clinical samples. In panel A, HMGS1 expression was studied in a clinical cohort, but I did not find information about the cohort. Was a H-score established because the staining appears very heterogenous. The absence of annotation on the section (normal cells, lymphatic vessel, cancer cell...) makes the understanding of the panel difficult. We are not all pathologist. Regarding Figure 5B, the association with tumor size is rather convincing but TMN staging is rather weak essentially with T2 tumors. Did the author quantified lymph-invasion in these sample and did they made association study with DFS. In panel C and in page 19 line 11, expression of HMGS1 was studied in lymph node metastasis, but again I did not find the data that led to the conclusion and the statistics, the lack of annotation on

panel C is also a problem because the staining is heterogenous.

9- Finally, on figure 5D, the author studied both CD44 expression as a marker of stemness and SREBP2 as a marker of cholesterol metabolism on two tumor samples and again I do not understand the images.

Reviewer #2 (Comments to the Authors (Required)):

General comments: This is a very well written paper describing the involvement of snoRNA in the metastatic process by altering lipid metabolism and shifting the metastatic cells metabolic pathway toward the mevalonate pathway. The conclusion is supported by high quality well controlled in vitro and clinical data that will be of interest to a broad spectrum of basic and clinical investigators.

Specific comments:

Fig. 1D, 2B-C: Should also show DKC1 down regulation at the protein level.

Fig.1E: Only two samples of each representative groups are shown. In order to evaluate the significance of the percentage described in the text, the authors should report the number of samples analyzed from each cohort.

Fig.3E: The number of colonies not just the size should be provided .In addition, the shControl should be provided as well to rule out the effect of Doxycyclin.

Fig.5D: The description of these results is rather subjective and should be supported by quantifiable results.

Typo: p.14 line 20: "through diffusion"

Answers to Reviewer 1

The additional files should be better organized in figure and table, in the text calling additional files instead of supplementary figure is confusing.

We are grateful to the Reviewers for the exhaustive work performed, and we apologize for the inconvenient organization. We have now improved the format of the manuscript following LSA recommendations for article organization. The supplementary information is now referred to as figures S1 and S2 and tables S1 and S2.

1. Page 3 line 9 DKC1 is not defined, line 14 HMGCS1 is not defined same in page 6 line 9

We have fixed these errors and checked for any other mistakes.

2. Material and methods, page 7 IHC the scoring method is not explained, did the author use a threshold for positivity?

Staining score for DKC1 and HMGCS1 was performed to classify samples as negative (negative, trace or low staining) or positive (medium or strong staining), as stated in Material and Methods, without any additional classification criteria. See also answer to point 5.

3. Results, page 12 lines 4-7, the authors should mention the cell line used in the text and in figure 1D and comment their choice. If I am correct, in Fig. 1D, the authors use SCC13 to show the link between reduction of DKC1 and snoRNAs expression. This is also substantiated in supplementary Figure 1 with SCC12 and UT-SCC12A cells. I understood that SCC12 and 13 were differentiated cSCC cell line, not representative of Metastatic M SCC. However, the effect of siDKC1 is similar than in the metastatic UT-SCC12A. Is there an explanation?

We have now included orderly information of the three cell lines used, as required by the reviewer. The explanation for the effect of DKC1 on snoRNA expression levels in the three cSCC cell lines relies on the formation of the H/ACA ribonucleoprotein (RNP) complexes, in which DKC1 binding is critical for snoRNA stability in many different cellular contexts, as already reported in oral cancer and osteosarcoma cells, for instance. Clarification on the cell lines used and the explanation for DKC1 effect in the three cSCC cell lines can now be found in page 8:

To investigate the role of DKC1 in snoRNA expression levels we initially chose the SCC13 cell line, originated from a well differentiated human SCC of the facial epidermis (Rheinwald & Beckett, 1981). Transient transfection of these cells with a small interference RNA (siRNA) directed to DKC1 (siDKC1) showed that DKC1 knockdown resulted in drops in H/ACA snoRNA levels (Fig 1D). This effect was also observed in the highly differentiated cell line SCC12 and in the UT-SCC12A cell line, which was established from a primary human cSCC that later on metastasized (Jääskelä-Saari et al, 1997) (Fig S1). These findings were consistent with previous data showing that loss of DKC1 reduces the accumulation of a subset of H/ACA snoRNA-derived miRNAs in different cellular contexts (Alawi & Lin, 2010)

and supported the possibility that down-regulation of DKC1 underlies the impoverishment of H/ACA snoRNAs in metastatic cSCC.

4- Page 11, the author did a validation in an independent cohort which reinforces their finding, could they comment on the selection of SnoRNA18?

- Although we initially considered that validation of one of the snoRNAs in an independent cohort proved the findings described, we have now also validated the reduced expression of the other three snoRNAs in MSCC, and these data have been incorporated in Fig 1C.

5. A clinical study on DKC1 expression was performed on tissue microarrays page 12 lines 7-13, I did not find the data leading to the conclusions. Figure 1E is an illustration of the microarray series I suppose? It should indicate that DKC1 is labelled, is there an H-score? The subpanels should be indicated and are Metastases from the same patient than MSCC? The scale bar is difficult to see.

- As indicated, DKC1 and HMGCS1 expression was assessed in tissue microarrays of primary tumors and metastases on the basis of negative/poor or medium/high expression. We discarded a quantification based on an histoscore as for this assessment not only the expression levels of these proteins but also the percentage of expressing cells should be assessed in larger areas of the tumors, rather than in representative samples of the microarrays.

- We have now included a table (Fig 1E) with DKC1 expression data in MSCC, NMSCC and metastasis. We have also included labeling for DKC1, indications for the subpanels, and a magnified scale bar. Figure legend for panel 1E is:

(E) Table summarizing immunohistochemical expression of DKC1 in TMAs of primary non-metastasizing and metastasizing cutaneous squamous cell carcinomas (NMSCC and MSCC respectively), and in metastases. Initial cohorts: 56 MSCC, 51 non-MSCC and 39 metastases. In some cases, signal could not be evaluated due to technical limitations. Below, representative images showing DKC1 immunodetection in independent TMA samples of NMSCC (a, b), MSCC (c, d) or lymph metastases (d, e). Scale bar, 50 μ m.

6. Regarding gene upregulated in the absence of DCK1, page 13, lines 13 it is stated that 82 were found. Then the meaning of lines 14-15 is very unclear, "42 beard catalytic activity and 45 were involved in acetylation processes". Beard means bore? had? Figure 2 A is difficult to understand the red and dots legend could be added directly on the figure. Line 19 ACSS2 is not defined. Figure 2G lacks size bars and is there quantification?

- This sentence has been corrected:

Page 9: Indeed, among them, 42 had catalytic activity and 45 were involved in acetylation processes.

- We have labelled the red dots with larger protein names.

- We have defined ACSS2, now in page 10:

Among these proteins were the enzyme ACS2 (Acyl-CoA Synthetase Short Chain Family Member 2), which catalyses the activation of acetate and produces acetyl-coenzyme A (acetyl-CoA) for use in energy generation and lipid synthesis, and HMGCS1, which facilitates the condensation of acetyl-CoA with acetoacetyl-CoA to form HMG-CoA, which is converted by HMG-CoA reductase (HMGCR) into mevalonate, a precursor for cholesterol synthesis.

- Size bars for figure 2G and quantification for LDL deposits have now been included. Legend for Figure 2G:

(G) LDL uptake in control and DKC1-depleted UT-SCC12A cells cultured with doxycycline for 3 days in basal conditions, analysed with BODIPY-labelled LDL (green) upon costaining with LC3 (red) and DAPI (blue), and detected by fluorescence microscopy as shown in the representative images. Scale bars, 50 μ m. On the right, quantification of LDL deposits analyzed with ImageJ. Student's T-test, **, $p < 0,005$; ***, $p < 0,001$.

7. Regarding migration and invasion effects, Figure 3, C, D and E lacks subpanel names and size bars and it seems that less cells were seeded in shDKC1? On page 19, lines3-4, the authors mention a link between simvastatin and cell migration but there are no data on cell migration and simvastatin treatment. However, on Figure 4C simvastatin seems to reduce DKC1 expression in cells not treated with Doxycycline.

- We have now included subpanel names and size bars for all the pictures. Regarding subpanel names, indications for them can be found in the figure legend:

(C) Phase-contrast imaging of control and shDKC1 cells, cultured in absence or presence of doxycycline for 3 days, and the same cultures stained with Calcein AM and visualized by fluorescence microscopy (pictures below). Scale bars, 50 μ m. (D) In vitro migration and invasion abilities of control and shDKC1 cells cultured with doxycycline for 3 days. Doxycycline-treated cells were seeded in serum-deprived medium in the top compartments, and migration or invasion through 0.1X Matrigel-coated inserts towards medium supplemented with 10 % fetal bovine serum (FBS) was quantified 24 hr later by Calcein AM staining. Lower pictures, initially seeded cells. Scale bars, 200 μ m. On the right, data obtained from two independent experiments performed in triplicate. Statistic performed by the Student's T-test. ***, $p < 0,001$.

- Although it appears that fewer cells were seeded in shDKC1, this visual effect could be due to the smaller cell size of DKC1-depleted cells. The same number of cells was initially seeded in all experimental conditions tested, and these experiments were performed several times and gave reproducible results.

- As required by the reviewer, we have performed in vitro experiments to analyze the effect of mevalonate pathway blockade on the invasiveness of DKC1-depleted cells. The data are now included in Figure 4D:

(D) Effect of simvastatin on the in vitro migration and invasion abilities of control and shDKC1 cells cultured with doxycycline, as indicated in Figure 3D. Data were obtained from

*two independent experiments performed in triplicate. Statistic performed by the Student's T-test. ***, $p < 0,001$; **, $p < 0,005$.*

- We agree with reviewer 1 in the observation that simvastatin appears to reduce DKC1 expression in non-doxycycline-treated cells. Although in-depth analysis of this effect is beyond the scope of our current research, the down-regulation of DKC1 by simvastatin is probably due to post-transcriptional mechanisms, as DKC1 mRNA is not affected by 1 or 2 μM simvastatin (data not shown). These mechanisms could be related to cellular feedback loops aimed at regulating altered cell metabolism, or to changes in proteins involved in DKC1 protein turnover. As this is an observation that we have not investigated sufficiently at the experimental level, speculative arguments about possible reasons have not been included in the text.

8. On page 19 and on Figure 5, the authors studied the relevance of their finding in clinical samples. In panel A, HMGS1 expression was studied in a clinical cohort, but I did not find information about the cohort. Was a H-score established because the staining appears very heterogenous. The absence of annotation on the section (normal cells, lymphatic vessel, cancer cell...) makes the understanding of the panel difficult. We are not all pathologist. Regarding Figure 5B, the association with tumor size is rather convincing but TMN staging is rather weak essentially with T2 tumors. Did the author quantified lymph-invasion in these sample and did they made association study with DFS. In panel C and in page 19 lane 11, expression of HMGS1 was studied in lymph node metastasis, but again I did not find the data that led to the conclusion and the statistics, the lack of annotation on panel C is also a problem because the staining is heterogenous.

- HMGS1 expression was determined, as was DKC1 expression, on tissue microarrays with samples from the clinical cohort indicated in the section Material and Methods. This cohort included fifty-six primary metastasizing cSCCs that had evolved to histologically confirmed lymph node (n=51) or cutaneous in-transit (n=5) metastases and a control group of fifty-one patients with cSCCs who had not developed any metastasis in a 5-year follow-up period were included in the study. Thirty-nine metastases (35 lymph node/4 cutaneous in-transit) from 39 patients were also evaluated. These samples had been histopathologically analyzed previously, and a detailed characterization of this cohort is referenced in the manuscript, as indicated in Material and Methods (Epithelial to mesenchymal transition markers are associated with an increased metastatic risk in primary cutaneous squamous cell carcinomas but are attenuated in lymph node metastases. *J Dermatol Sci* 72: 93–102.). As indicated in the manuscript, HMGS1 expression was restricted to some cells in invasive tumor areas, and the presence of positive cells was the criterion for considering a tumor positive for HMGS1 staining.

- Tissue sections shown in figure 5A mainly display tumoral cells. Following the reviewer recommendation, we have now labelled the different cell subtypes present in the sample sections, and we expect that this change will help to improve the interpretation of the images:

Figure 5. Expression of lipid metabolism markers in cSCC samples. A. HMGS1 expression in three primary cSCC samples (two different magnifications), detected by immunohistochemistry. Observe the intense cytoplasmic staining of some tumoral cells (blue arrows). Grey arrows, eccrine glands; asterisks, inflammatory infiltrate.

- Correlation analysis to investigate the association of HMGCS1 with TNM indicates that there is a bias towards higher stages among those tumors with HMGCS1 positive cells. In other words, although a similar number of HMGCS1-negative tumors are represented in stages T1 and T2 stages, HMGCS1-positive tumors are mostly found in stage T2. A similar trend could be observed using Brigham's criteon, but statistical significance was not reached probably due to the limited size of the cohort.

- We are currently performing immunohistochemical analysis to characterize the density of lymphatic vessels. However, reliable identification of lymphatic vessels in cSCC is technically complex. Numerous studies rely on the identification of such vessels in different tumors on the expression of podoplanin (D2-40), and the absence of expression of vessel-specific proteins, such as CD31 and CD105. However, while the exclusion of CD31 and CD105 expression is not accurate as lymphatic vessels may also express these proteins, the D2-40 staining frequently used to identify these structures in many other tumors is not adequate to identify lymphatic vessels in cSCC, as these tumors can express high D2-40 levels, hampering the detection of these vessels in up to 30% cases, as previously described by our group (J Am Acad Dermatol. 2012 Dec;67(6):1310-8. D2-40 immunohistochemical overexpression in cutaneous squamous cell carcinomas: a marker of metastatic risk). Given such difficulties, we are investigating the expression of other possible lymphatic vessel markers in consecutive whole tumor sections, as they allow to follow the histological structure of the vessel, but not in TMA tumor samples as they contain only a few millimeters of tumor tissue. All these considerations and limitations have prevented an accurate quantification of lymphatic vessel density and its association with DFS in our patients.

- Data for HMGCS1 expression in primary and metastatic samples has been included in panel 5C, and tumoral cells are now indicated by arrows:

C. Contingency table of HMGCS1 expression in primary cSCC and metastases. Pictures below, HMGCS1 expression in two representative lymph metastases at two different magnifications, detected by immunohistochemistry. Tumoral cells, blue arrows; lymphoid tissue, purple arrows.

9- Finally, on figure 5D, the author studied both CD44 expression as a marker of stemness and SREBP2 as a marker of cholesterol metabolism on two tumor samples and again I do not understand the images.

- We apologize for the poor explanation of this figure. We believe that the re-identification of the regions corresponding to these two cSCCs and the labelling of the cell subtypes have helped to considerably improve the quality of this panel. Figure legend now states:

D. Expression of CD44 (brown, membrane) and SREBP2 (pink, cytoplasm) in four different fields of two primary cSCC tumors. The basal layer of normal epidermis is outlined with a dashed line. Tumoral cells, blue arrows; asterisks, inflammatory infiltrate; Kp, keratin pearls. Pictures on the right, high magnification of fields in the same cases. Observe the presence of scarce cells with concomitant expression of high SREBP2 and CD44 levels. Scale bars, 100 μ m.

Answers to Reviewer 2

Specific comments:

Fig. 1D, 2B-C: Should also show DKC1 down regulation at the protein level.

The decrease in DKC1 protein levels upon siRNA transfection of the three cSCC cell lines is shown Figure S2A. We regret that we are unable to insert them between the main figures due to spatial constraints.

Fig.1E: Only two samples of each representative groups are shown. In order to evaluate the significance of the percentage described in the text, the authors should report the number of samples analyzed from each cohort.

As requested also by Reviewer 1, we have now included the table showing DKC1 expression data in each cohort (MSCC, NMSCC and metastasis). Figure legend for panel 1E is:

(E) Table summarizing immunohistochemical expression of DKC1 in TMAs of primary non-metastasizing and metastasizing cutaneous squamous cell carcinomas (NMSCC and MSCC respectively), and in metastases. Initial cohorts: 56 MSCC, 51 non-MSCC and 39 metastases. In some cases, signal could not be evaluated due to technical limitations. Below, representative images showing DKC1 immunodetection in independent TMA samples of NMSCC (a, b), MSCC (c, d) or lymph metastases (d, e). Scale bar, 50 μ m.

Fig.3E: The number of colonies not just the size should be provided. In addition, the shControl should be provided as well to rule out the effect of Doxycyclin.

We would like to apologize for the incorrect use of the term colony. This error has been corrected in the text, and has been replaced by the term aggregate. Indeed, for the study of stemness and clonogenicity of DKC1-depleted cells we performed a spheroid formation assays, a 3D assay that may better approximate in vivo biology than standard 2D colony formation assays. Specifically, we carried out the hanging drop method of 3D cell culture, which mimics the environment of the cells within a tissue. This method is preferred compared to the other 3D cell culture settings since it allows the cells to form spheroids, in which cells are cultured in close contact to each other, forming an aggregated cluster of cells cultured in 3D physiological condition. This method is widely used as a model for the study of cell-cell interactions, tumor tissue organization and microenvironment, and to test cell stemness. In this method, each droplet contains a single aggregate of cells, and all droplets have an aggregate.

To address the reviewer's concerns about doxycycline, we have included data from spheroids generated by doxycycline-treated shControl cells, as requested. The figure legend now reads:

(E) Images from one representative experiment out of two showing spheroids formed 7 days after plating *shControl* and *shDKC1 UT-SCC12A* cells with vehicle (-DOX) or with doxycycline (+DOX) (left pictures) (scale bars, 200 μm). On the right, graph showing the quantification of the spheroid areas (arbitrary units, a.u.) (left). Mean + s.e.m., two-tailed t-test.

Fig.5D: The description of these results is rather subjective and should be supported by quantifiable results.

The sole purpose of CD44/HMGCS1 double labelling was to determine the expression of these two proteins in different areas of the tumor and to investigate the possible presence of double-positive cells. Therefore, we show images of such expression in two skin biopsies of cSCC containing healthy skin adjacent to the tumor and two representative regions, one more superficial and one more infiltrative. These pictures illustrate the progressive acquisition of expression of both proteins as the tumor invades the underlying layers. In addition, this labelling allowed us to identify cells with exacerbated lipid metabolism and high expression of the stem cell marker CD44 in deep regions of the tumor. Therefore, although we appreciate that a quantitative assessment could be interesting from a diagnostic point of view, such an analysis requires a complete biopsy of each tumor, which we do not have at our disposal. In any case, we apologize for the poor explanation of this figure, which may have led reviewer 2 to confusion about the purpose of this panel. We believe that the re-identification of the regions corresponding to these two cSCCs and the labelling of the cellular subtypes will help to better understand this panel. The figure legend now reads:

D. Expression of CD44 (brown, membrane) and SREBP2 (pink, cytoplasm) in four different fields of two primary cSCC tumors. The basal layer of normal epidermis is outlined with a dashed line. Tumoral cells, blue arrows; asterisks, inflammatory infiltrate; Kp, keratin pearls. Pictures on the right, high magnification of fields in the same cases. Observe the presence of scarce cells with concomitant expression of high SREBP2 and CD44 levels. Scale bars, 100 μm .

Typo: p.14 line 20: "through diffusion".

We have corrected this typographical error, and would like to acknowledge the detailed revision of the manuscript by reviewer 2.

January 12, 2023

RE: Life Science Alliance Manuscript #LSA-2022-01692-TR

Dr. Inma Hernandez-Munoz
Institut Hospital del Mar d'Investigacions Mèdiques
Group of Inflammatory and Neoplastic Dermatological Diseases
Doctor Aiguader, 88
Barcelona 8003
Spain

Dear Dr. Hernandez-Munoz,

Thank you for submitting your revised manuscript entitled "LOSS OF DYSKERIN FACILITATES THE ACQUISITION OF METASTATIC TRAITS BY ALTERING THE MEVALONATE PATHWAY". We would be happy to publish your paper in Life Science Alliance pending final revisions necessary to meet our formatting guidelines.

- please add ORCID ID for corresponding author-you should have received instructions on how to do so
- please add the Twitter handle of your host institute/organization as well as your own or/and one of the authors in our system
- please make GEO accession GSE207545 and PRIDE dataset PXD034155 publicly accessible at this point

Figure Check:

- please add sizes next to all blots

A. FINAL FILES:

B. MANUSCRIPT ORGANIZATION AND FORMATTING:

Sincerely,

Reviewer #1 (Comments to the Authors (Required)):

The authors have addressed my concerns

Reviewer #2 (Comments to the Authors (Required)):

This is a very well written paper describing the involvement of snoRNA in the metastatic process by altering lipid metabolism and shifting the metastatic cells metabolic pathway toward the mevalonate pathway. The conclusion is supported by high quality well controlled in vitro and clinical data that will be of interest to a broad spectrum of basic and clinical investigators. All comments previously raised by this reviewer were properly addressed in the revised manuscript.

January 19, 2023

RE: Life Science Alliance Manuscript #LSA-2022-01692-TRR

Dr. Inma Hernandez-Munoz
Institut Hospital del Mar d'Investigacions Mèdiques
Group of Inflammatory and Neoplastic Dermatological Diseases
Doctor Aiguader, 88
Barcelona 8003
Spain

Dear Dr. Hernandez-Munoz,

Thank you for submitting your Research Article entitled "LOSS OF DYSKERIN FACILITATES THE ACQUISITION OF METASTATIC TRAITS BY ALTERING THE MEVALONATE PATHWAY". It is a pleasure to let you know that your manuscript is now accepted for publication in Life Science Alliance. Congratulations on this interesting work.

DISTRIBUTION OF MATERIALS:

Again, congratulations on a very nice paper. I hope you found the review process to be constructive and are pleased with how the manuscript was handled editorially. We look forward to future exciting submissions from your lab.

Sincerely,
